# GeneOH Diffusion: Towards Generalizable Hand-Object Interaction Denoising via Denoising Diffusion

**Xueyi Liu**[1,3]    **Li Yi**[1,2,3]
[1]Tsinghua University  [2]Shanghai AI Laboratory  [3]Shanghai Qi Zhi Institute
Project website: meowuu7.github.io/GeneOH-Diffusion

## Abstract

In this work, we tackle the challenging problem of denoising hand-object interactions (HOI). Given an erroneous interaction sequence, the objective is to refine the incorrect hand trajectory to remove interaction artifacts for a perceptually realistic sequence. This challenge involves intricate interaction noise, including unnatural hand poses and incorrect hand-object relations, alongside the necessity for robust generalization to new interactions and diverse noise patterns. We tackle those challenges through a novel approach, **GeneOH Diffusion**, incorporating two key designs: an innovative contact-centric HOI representation named GeneOH and a new domain-generalizable denoising scheme. The contact-centric representation GeneOH informatively parameterizes the HOI process, facilitating enhanced generalization across various HOI scenarios. The new denoising scheme consists of a canonical denoising model trained to project noisy data samples from a whitened noise space to a clean data manifold and a "denoising via diffusion" strategy which can handle input trajectories with various noise patterns by first diffusing them to align with the whitened noise space and cleaning via the canonical denoiser. Extensive experiments on four benchmarks with significant domain variations demonstrate the superior effectiveness of our method. GeneOH Diffusion also shows promise for various downstream applications.

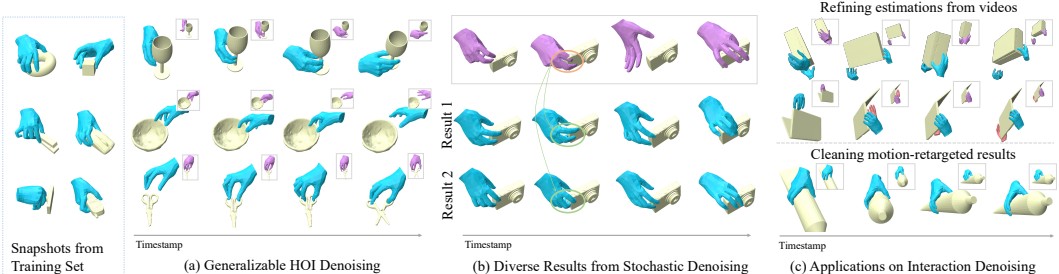

Figure 1: Trained only on limited data, **GeneOH Diffusion** can clean novel noisy interactions with new objects, hand motions, and unseen noise patterns (*Fig. (a)*), produces diverse refined trajectories with discrete manipulation modes (*Fig. (b)*), and is a practical tool for many applications (*Fig. (c)*).

## 1 Introduction

Interacting with objects is an essential part of our daily lives, and accurately tracking hands during these interactions has become crucial for various applications, such as gaming, virtual and augmented reality, robotics, and human-machine interaction. Yet, this task is highly complex and ill-posed due to numerous factors like intricate dynamics involved and hand-object occlusions. Despite best efforts, existing tracking algorithms often struggle with producing plausible and realistic results.

To better cater to the requirements of downstream tasks, noisy tracking results usually need to be refined. Given a hand-object interaction (HOI) sequence with errors, the HOI denoising aims to pro-

duce a natural interaction sequence free of artifacts such as penetrations. In this work, we assume the object poses are tracked accurately and focus on refining the hand trajectory following (Zhou et al., 2022; Grady et al., 2021; Zhou et al., 2021b; Zhang et al., 2021). This setting is important with many practical demands in applications such as cleaning synthesized motions (Tendulkar et al., 2023; Huang et al., 2023; Ghosh et al., 2023; Wu et al., 2022), refining motion-retargeted trajectories (Hecker et al., 2008; Tak & Ko, 2005; Aberman et al., 2019), and virtual object manipulations (Oh et al., 2019; Kato et al., 2000; Shaer et al., 2010). Early approaches relied on manually designed priors (Dewaele et al., 2004; Hackenberg et al., 2011), which, however, proved inadequate in handling intricate noise. More recent endeavors have shifted towards learning denoising priors from data (Zhou et al., 2022; 2021b; Grady et al., 2021), yet the existing designs still fall short of providing a satisfactory solution.

Leveraging data priors for HOI denoising is challenged by several difficulties. First, the interaction noise is highly complex, covering unnatural hand poses, erroneous hand-object spatial relations, and inconsistent hand-object temporal relations. Second, hand movements, hand-object relations, and the noise pattern may vary dramatically across different HOI tracks. For instance, the noise pattern exhibited in hand trajectories estimated from videos differs markedly from that resulted from inaccurate capturing or annotations. A denoising model is often confronted with such out-of-domain data and is expected to handle them adeptly. However, such a distribution shift poses a substantial challenge for data-driven models. Lacking an effective solution, prior works always cannot clean such complex interaction noise or can hardly generalize to unseen erroneous interactions.

We propose **GeneOH Diffusion**, a powerful denoising method with strong generalizability and practical applicability (see Figure 1), to tackle the above difficulties. Our method resolves the challenges around two key ideas: 1) designing an effective HOI representation that can both informatively parameterize the interaction and facilitate the generalization by encoding and canonicalizing vital HOI information in a coordinate system induced by the interaction region; 2) learning a canonical denoiser that projects noisy data from a whitened noise space to the data manifold for domain-generalizable denoising. A satisfactory representation that parameterizes the high-dimensional HOI process for denoising should be able to represent the interaction process faithfully, highlight noises, and align different HOI tracks well to enhance generalization capabilities Therefore, we introduce **GeneOH**, **Gene**ralized contact-centric **H**and-**O**bject spatial and temporal relations. GeneOH encodes the interaction informatively, encompassing the hand trajectory, hand-object spatial relations, and hand-object temporal relations. Furthermore, it adopts a contact-centric perspective and incorporates an innovative canonicalization strategy. This approach effectively reduces disparities between different sequences, promoting generalization across diverse HOI scenarios. To enhance the denoising model's generalization ability to novel noise distributions, our second effort centers on the denoising scheme side. We propose to learn a canonical denoising model that describes the mapping from a whitened noise space to the data manifold. The whitened noise space contains noisy data diffused from clean data in the training dataset via Gaussian noise at various noise scales. With the canonical denoiser, we then leverage a "denoising via diffusion" strategy to handle input trajectories with various noise patterns in a domain-generalizable manner. It first aligns the input to the whitened noise space by diffusing it via Gaussian noise. Subsequently, the diffused sample is cleaned by the canonical denoising model. To strike a balance between the denoising model's generalization capability and the faithfulness of the denoised trajectory, we introduce a hyper-parameter that decides the scale of noise added during the diffusion process, ensuring the diffused sample remains faithful to the original input. Furthermore, instead of learning to clean the interaction noise through a single stage, we devise a progressive denoising strategy where the input is sequentially refined via three stages, each of which concentrates on cleaning one specific component of GeneOH .

We conduct extensive experiments on three datasets, GRAB (Taheri et al., 2020), a high-quality MoCap dataset, HOI4D (Liu et al., 2022), a real-world interaction dataset with noise resulting from inaccurate depth sensing and imprecise vision estimations, and ARCTIC (Fan et al., 2023), a dataset featuring dynamic motions and changing contacts, showing the remarkable effectiveness and generalizability of our method. When only trained on GRAB, our denoiser can generalize to HOI4D with novel and difficult noise patterns and ARCTIC with challenging interactions, surpassing prior arts by a significant margin, as demonstrated by the comprehensive quantitative and qualitative comparisons. We will release our code to support future research. In summary, our contributions include:

- An HOI denoising framework with powerful spatial and temporal denoising capability and unprecedented generalizability to novel HOI scenarios;

- An HOI representation named GeneOH that can faithfully capture the HOI process, highlight unnatural artifacts, and align HOI tracks across different objects and interactions;
- An effective and domain-generalizable denoising method that can both generalize across different noise patterns and clean complex noise through a progressive denoising strategy.

## 2 RELATED WORKS

Hand-object interaction is an important topic for understanding human behaviors. Prior works towards this direction mainly focus on data collection (Taheri et al., 2020; Hampali et al., 2020; Guzov et al., 2022; Fan et al., 2023; Kwon et al., 2021), reconstruction (Tiwari et al., 2022; Xie et al., 2022; Qu et al., 2023; Ye et al., 2023), interaction generation (Wu et al., 2022; Tendulkar et al., 2023; Zhang & Tang, 2022; Ghosh et al., 2023; Li et al., 2023), and motion refinement (Zhou et al., 2022; Grady et al., 2021; Zhou et al., 2021b; Núñez, 2022). The HOI denoising task wishes to remove unnatural phenomena from HOI sequences with interaction noise. In real application scenarios, a denoising model would frequently encounter out-of-domain interactions, and is expected to generalize to them. This problem is then related to domain generalization, a general machine learning topic (Sicilia et al., 2023; Segu et al., 2023; Wang et al., 2023; Zhang et al., 2023; Jiang et al., 2022; Wang et al., 2022; Blanchard et al., 2011; Muandet et al., 2013; Dou et al., 2019), where a wide range of solutions have been proposed in the literature. Among them, leveraging domain invariance to solve the problem is a promising solution. Our work is related to this kind of approach, at a high level. However, what is the domain invariant information for the HOI denoising task, and how to encourage the model to leverage such information for denoising remains very tricky. We focus on designing invariant representations and learning a canonical denoiser for domain-generalizable denoising. Moreover, we are also related to intriguing works that wish to leverage data priors to solve the inverse problem (Song et al., 2023; Mardani et al., 2023; Tumanyan et al., 2023; Meng et al., 2021; Chung et al., 2022). For our task, we need to answer some fundamental questions regarding what are generalizable denoising priors, how to learn them from data, and how to leverage the prior to refine noisy input from different distributions. We'll illustrate our solution in the method section.

## 3 HAND-OBJECT INTERACTION DENOISING VIA DENOISING DIFFUSION

Given an erroneous hand-object interaction sequence with $K$ frames $(\hat{\mathcal{H}}, \mathbf{O}) = \{(\hat{\mathbf{H}}_k, \mathbf{O}_k)\}_{k=1}^{K}$, we assume the object pose trajectory $\{\mathbf{O}_k\}_{k=1}^{K}$ is accurate following (Zhou et al., 2022; 2021b; Grady et al., 2021; Zhang et al., 2021) and aim at cleaning the noisy hand trajectory $\{\hat{\mathbf{H}}_k\}_{k=1}^{K}$. This setting is of considerable importance, given its practical applicability in various domains (Tendulkar et al., 2023; Ghosh et al., 2023; Li et al., 2023; Wu et al., 2022; Hecker et al., 2008; Oh et al., 2019; Shaer et al., 2010). The cleaned hand trajectory should be free of unnatural hand poses, incorrect spatial penetrations, and inconsistent temporal hand-object relations. The hand trajectory should present visually consistent motions and adequate contact with the object to support manipulation. The problem is ill-posed in nature owing to the difficulties posed by complex interaction noise and the substantial domain gap across different interactions resulting from new objects, hand movements, and unseen noise patterns.

We resolve the above difficulties by 1) designing a novel HOI representation that parameterizes the HOI process faithfully and can both simplify the distribution of complex HOI and foster the model generalization across different interactions (Section 3.1) and 2) devising an effective denoising scheme that can both clean complex noises through a progressive denoising strategy and generalize across different input noise patterns (Section 3.2).

### 3.1 GENEOH : GENERALIZED CONTACT-CENTRIC HAND-OBJECT SPATIAL AND TEMPORAL RELATIONS

Designing an effective and generalizable HOI denoising model requires a serious effort in the representation design. It involves striking a balance between expressive modeling of the interaction with objects and supporting the model's generalization to new objects and interactions. The ideal HOI representation should accurately capture the interaction process, highlight any unusual phenomena like spatial penetrations, and facilitate alignment across diverse interaction sequences.

We introduce GeneOH to achieve this. It integrates the hand trajectory, hand-object spatial relations, and hand-object temporal relations to represent the HOI process faithfully. An effective normalization strategy is further introduced to enhance alignment across diverse interactions. The hand trajectory and the object trajectory are compactly represented as the trajectory of hand keypoints, denoted as $\mathcal{J} = \{\mathbf{J}_k\}_{k=1}^K$, and the interaction region sequence: $\mathcal{P} = \{\mathbf{P}_k\}_{k=1}^K$, in a contact-aware manner. We will then detail the design of GeneOH .

**Generalized contact points.** The interaction region is established based on points sampled from the object surface close to the hand trajectory, referred to as "generalized contact points". They are $N_o$ points (denoted as $\mathbf{P} \in \mathbb{R}^{N_o \times 3}$) sampled from object surface points, whose distance to the hand trajectory does not exceed a threshold value of $r_c$ (set to 5mm). The sequence of these points across all frames is represented by $\mathcal{P} = \{\mathbf{P}_k\}_{k=1}^K$, where $\mathbf{P}_k$ denotes points at frame $k$. Each $\mathbf{P}_k$ is associated with a 6D pose, consisting of the object's orientation (or the orientation of the first part for articulated objects), denoted as $\mathbf{R}_k$, and the center of $\mathbf{P}_k$, denoted as $\mathbf{t}_k$.

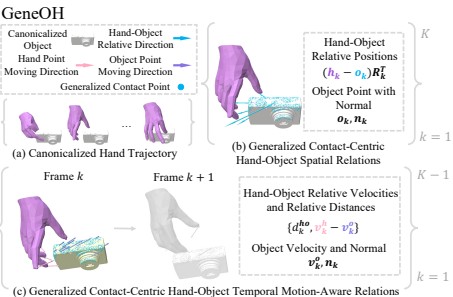

Figure 2: Three components of **GeneOH**.

**Canonicalized hand trajectories.** We include hand trajectories in our representation to effectively model hand movements. Specifically, we leverage hand keypoints to model the hand, as they offer a compact and expressive representation. We represent the hand trajectory as the sequence of 21 hand keypoints, denoted as $\mathcal{J} = \{\mathbf{J}_k \in \mathbb{R}^{N_h \times 3}\}_{k=1}^K$, where $N_h = 21$. We further canonicalize the hand trajectory $\mathcal{J}$ using the poses of the generalized contact points to eliminate the influence of object poses, resulting in the canonicalized hand trajectory in GeneOH : $\bar{\mathcal{J}} = \{\bar{\mathbf{J}}_k = (\mathbf{J}_k - \mathbf{t}_k)\mathbf{R}_k^T\}_{k=1}^K$.

**Generalized contact-centric hand-object spatial relations.** We further introduce a hand-object spatial representation in GeneOH. The representation is based on hand keypoints and generalized contact points to inherit their merits. The spatial relation centered at each generalized contact point $\mathbf{o}_k \in \mathbf{P}_k$ comprises the relative offset from $\mathbf{o}_k$ to each hand keypoint $\mathbf{h}_k \in \mathbf{J}_k$, *i.e.*, $\{\mathbf{h}_k - \mathbf{o}_k | \mathbf{h}_k \in \mathbf{J}_k\}$, the object point normal $\mathbf{n}_k$, and the object point position $\mathbf{o}_k$. These statistics are subsequently canonicalized using the 6D pose of the generalized contact points to encourage cross-interaction alignment. Formally, the spatial representation centered at $\mathbf{o}_k$ is defined as: $\mathbf{s}_k^{\mathbf{o}} = ((\mathbf{o}_k - \mathbf{t}_k)\mathbf{R}_k^T, \mathbf{n}_k\mathbf{R}_k^T, \{(\mathbf{h}_k - \mathbf{o}_k)\mathbf{R}_k^T | \mathbf{h}_k \in \mathbf{J}_k\})$. The spatial relation $\mathcal{S}$ is composed of $\mathbf{s}_k^{\mathbf{o}}$ at each generalized contact point: $\mathcal{S} = \{\{\mathbf{s}_k^{\mathbf{o}} | \mathbf{o}_k \in \mathbf{P}_k\}\}_{k=1}^K$. By encoding object normals and hand-object relative offsets, $\mathcal{S}$ can reveal unnatural hand-object spatial relations such as penetrations.

**Generalized contact-centric hand-object temporal relations.** Considering the limitations of the above two representations in revealing temporal errors such as incorrect manipulations resulting from inconsistent hand-object motions, we further introduce hand-object temporal relations to parameterize the HOI temporal information explicitly. We again take hand keypoints $\mathbf{J}$ to represent hand shape and generalized contact points $\mathbf{P}$ for the object shape to take advantage of their good ability in supporting generalization. The temporal relations encode the relative velocity between each hand point $\mathbf{o}_k$ and each hand keypoint $\mathbf{h}_k$ at frame $k$ ($\mathbf{v}_k^{\mathbf{ho}} = \mathbf{v}_k^{\mathbf{h}} - \mathbf{v}_k^{\mathbf{o}}$), the Euclidean distance between each pair of points ($d_k^{\mathbf{ho}} = \|\mathbf{h}_k - \mathbf{o}_k\|_2$), and the object velocity $\mathbf{v}_k^o$ in the representation, as illustrated in Figure 2. We further introduce two statistics by using the object point normal to canonicalize $\mathbf{v}_k^{\mathbf{ho}}$, resulting in two normalized statistics: $\mathbf{v}_{k,\perp}^{\mathbf{ho}}$, orthogonal to the object tangent plane, and $\mathbf{v}_{k,\parallel}^{\mathbf{ho}}$, lying in the object's tangent plane, and encoding them with hand-object relative distances: $e_{k,\perp}^{\mathbf{ho}} = e^{-k \cdot d_k^{\mathbf{ho}}} k_b \|\mathbf{v}_{k,\perp}^{\mathbf{ho}}\|_2$ and $e_{k,\parallel}^{\mathbf{ho}} = e^{-k \cdot d_k^{\mathbf{ho}}} k_a \|\mathbf{v}_{k,\parallel}^{\mathbf{ho}}\|_2$. Here, $k$, $k_a$, and $k_b$ are positive hyperparameters, and the term $e^{-k \cdot d_k^{\mathbf{ho}}}$ is negatively related to the distance between the hand and object points. This canonicalization and encoding strategy aims to encourage the model to learn different denoising strategies for the two types of relative velocities, enhance cross-interaction generalization by factoring out object poses, and emphasize the relative movement between very close hand-object point pairs. The temporal representation $\mathcal{T}$ is defined by combining the above statistics of each hand-object point pair across all frames together:

$$\mathcal{T} = \{\{\mathbf{v}_k^{\mathbf{o}}, \{d_k^{\mathbf{ho}}, \mathbf{v}_k^{\mathbf{ho}}, e_{k,\parallel}^{\mathbf{ho}}, e_{k,\perp}^{\mathbf{ho}} | \mathbf{h}_k \in \mathbf{J}_k\}\} | \mathbf{o}_k \in \mathbf{P}_k\}_{k=1}^{K-1}. \tag{1}$$

It reveals temporal errors by encoding object velocities, hand-object distances and relative velocities.

Figure 3: The **progressive HOI denoosing** gradually cleans the input noisy trajectory through three stages. Each stage concentrates on refining the trajectory by denoising a specific part of GeneOH via a *canonical denoiser* through the *"denoising via diffusion"* strategy.

**The GeneOH representation.** The overall representation, GeneOH, comprises the above three components, as defined formally: GeneOH $= \{\bar{\mathcal{J}}, \mathcal{S}, \mathcal{T}\}$. Figure 2 illustrates the design. It faithfully captures the interaction process, can reveal noise by encoding corresponding statistics, and benefits the generalization by employing carefully designed canonicalization strategies. Inspecting back into previous works, TOCH (Zhou et al., 2022) does not explicitly parameterize the hand-object temporal relations or hand shapes and does not carefully consider the spatial canonicalization to facilitate the generalization, which limits its denoising capability and may lead to the loss of high-frequency hand pose details. ManipNet (Zhang et al., 2021) does not encode temporal relations and does not incorporate contact-centric canonicalization, rendering it inadequate for capturing the interaction process and less effective for generalization purposes.

## 3.2 GeneOH Diffusion: Progressive HOI Denoising via Denoising Diffusion

While GeneOH excels in encoding the interaction process faithfully, highlighting errors to facilitate denoising, and reducing the disparities among various interaction sequences, designing an effective denoising model is still challenged by complex interaction noise, even from a distribution unseen during training. Previous methods typically employ pattern-specific denoising models trained to map noisy data restricted to certain patterns to the clean data manifold (Zhou et al., 2022; 2021b). However, these methods are susceptible to overfitting, resulting in conceptually incorrect results when faced with interactions with unseen noise patterns, as evidenced in our experiments.

---

**Algorithm 1 Denoising via Diffusion**

---

**Input:** forward diffusion function Diffuse$(\cdot, t)$, the denoising model denoise$(\cdot, t)$, input noisy point $\hat{x}$, diffusion steps $t_{\text{diff}}$.
**Output:** denoised data $x$.
  1: **function** DENOISE($\tilde{x}^{t_{\text{diff}}}, t_{\text{diff}}$)
  2:     **for** $t$ from $t_{\text{diff}}$ to 1 **do**
  3:         $\tilde{x}^{t-1} \sim$ denoise$(\tilde{x}^t, t)$
  4:   **return** $\tilde{x}^0$
  5: $\tilde{x} \leftarrow$ Diffuse$(\hat{x}, t_{\text{diff}})$
  6: **return** $x \leftarrow$ DENOISE$(\tilde{x}, t_{\text{diff}})$

---

To ease the challenge posed by novel interaction noise, we propose a new denoising paradigm that learns a canonical denoising model and leverages it for domain-generalizable denoising. It describes the mapping from noisy data at various noise scales from a whitened noise space to the data manifold. The whitened noise space is populated with noisy data samples diffused from the clean data via a *diffusion process* which gradually adds Gaussian noise to the data according to a variance schedule, a similar flavor to the forward diffusion process in diffusion-based generative models (Song et al., 2020; Ho et al., 2020; Rombach et al., 2022; Dhariwal & Nichol, 2021). With the canonical denoiser, we then leverage a "denoising via diffusion" strategy to handle input trajectories with various noise patterns in a generalizable manner. It first diffuses the input trajectory $\hat{x}$ via the diffusion process to another sample $\tilde{x}$ that resides closer to the whitened noise space. Then the model projects the diffused sample $\tilde{x}$ to the data manifold. To balance the generalization ability of the denoising and the fidelity of the denoised result to the input, the diffused $\tilde{x}$ needs to be faithful to the input $\hat{x}$. We then introduce a diffusion timestep $t_{\text{diff}}$ that decides how many diffusion steps are added. The process is visually depicted in the right part of Figure 3. Details are outlined in Algorithm 1. We also implement the denoising model's function and the training as those of the score functions in diffusion-based generative models. It is a multi-step stochastic denoiser that eliminates the noise of the input gradually to zero step-by-step. This way the denoiser can deal with noise at different scales flexibly and can give multiple solutions for the ill-posed ambiguous denoising problem.

Based on the domain-generalizable denoising strategy, designing a single data-driven model to clean heterogeneous interaction noise in one stage is still not feasible. The interaction noise contains various kinds of noise at ununiform scales stemming from different reasons. Thus the corresponding

noise-to-data mapping is very high dimensional and is very challenging to learn from limited data. A promising solution to tackle the complexity is taking a progressive approach and learning multiple specialists, each concentrating on cleaning a specific type of noisy information. However, the multi-stage formulation brings new difficulties. It necessitates careful consideration of the information to be cleaned at each stage to prevent the current stage from compromising the naturalness achieved in previous stages. Fortunately, our design of the GeneOH representation facilitates a solution to this issue. HOI information can be represented into three relatively homogeneous parts: $\bar{\mathcal{J}}$, $\mathcal{S}$, and $\mathcal{T}$. Furthermore, their relations ensure the sequential refinement of the hand trajectory by denoising its $\bar{\mathcal{J}}$, $\mathcal{S}$, and $\mathcal{T}$ representations across three stages can avoid the undermining problem. A formal proof of this property is provided in the Appendix A.2.

**Progressive HOI denoising.** We design a three-stage denoising approach (outlined in Figure 3), each stage dedicated to cleaning one aspect of the representation: $\bar{\mathcal{J}}$, $\mathcal{S}$, and $\mathcal{T}$, respectively. In each stage, a canonical denoising model is learned for the corresponding representation, and the denoising is carried out using the "denoising via diffusion" strategy. Given the input $\text{GeneOH}^{\text{input}} = \{\hat{\bar{\mathcal{J}}}^{\text{input}}, \hat{\mathcal{S}}^{\text{input}}, \hat{\mathcal{T}}^{\text{input}}\}$, the first denoising stage, named **MotionDiff**, denoises the noisy canonical hand trajectory $\hat{\bar{\mathcal{J}}}^{\text{input}}$ to $\bar{\mathcal{J}}^{\text{stage}_1}$. One stage-denoised hand trajectory $\mathcal{J}^{\text{stage}_1}$ can be easily computed by de-canonicalizing $\bar{\mathcal{J}}^{\text{stage}_1}$ using object poses. $\text{GeneOH}^{\text{input}}$ can also be updated accordingly into $\text{GeneOH}^{\text{stage}_1} = \{\bar{\mathcal{J}}^{\text{stage}_1}, \hat{\mathcal{S}}^{\text{stage}_1}, \hat{\mathcal{T}}^{\text{stage}_1}\}$. Then the second stage, named **SpatialDiff**, denoises the noisy spatial relation $\hat{\mathcal{S}}^{\text{stage}_1}$ to $\mathcal{S}^{\text{stage}_2}$. Two stages-denoised hand trajectory $\mathcal{J}^{\text{stage}_2}$ can be transformed from the hand-object relative offsets in $\mathcal{S}^{\text{stage}_2}$: $\mathcal{J}^{\text{stage}_2} = \text{Average}\{(\mathbf{h}_k - \mathbf{o}_k) + \mathbf{o}_k | \mathbf{o}_k \in \mathbf{P}_k\}$. Following this, $\text{GeneOH}^{\text{stage}_1}$ will be updated to $\text{GeneOH}^{\text{stage}_2} = \{\bar{\mathcal{J}}^{\text{stage}_2}, \mathcal{S}^{\text{stage}_2}, \hat{\mathcal{T}}^{\text{stage}_2}\}$. Finally the last stage, named **TemporalDiff**, denoises $\hat{\mathcal{T}}^{\text{stage}_2}$ to $\mathcal{T}^{\text{stage}_3}$. Since temporal information such as relative velocities is redundantly encoded in $\mathcal{T}$, we compute the three stages-denoised hand trajectory $\mathcal{J}^{\text{stage}_3}$ by optimizing $\mathcal{J}^{\text{stage}_2}$ so that its induced temporal representation aligns with $\mathcal{T}^{\text{stage}_3}$. And we take $\mathcal{J}^{\text{stage}_3}$ as the final denoising output, denoted as $\mathcal{J}$. Each stage would not undermine the naturalness achieved after the previous stages, as proved in the Appendix A.2.

**Fitting for a hand mesh trajectory.** With the denoised trajectory $\mathcal{J}$ and the object trajectory, a parameterized hand sequence represented via MANO parameters $\{\mathbf{r}_k, \mathbf{t}_k, \beta_k, \theta_k\}_{k=1}^{K}$ are optimized to fit $\mathcal{J}$ well. Details are illustrated in the Appendix A.3.

# 4 EXPERIMENTS

We conduct extensive experiments to demonstrate the effectiveness of our method. We train all models on the same training dataset and introduce four four test sets with different levels of domain shift to assess their denoising ability and the generalization ability (see Section 4.2). Moreover, we demonstrate the ability of our denoising method to produce multiple reasonable solutions for a single input in Section 4.3. At last, we show various applications that we can support (Section 4.4). *Another series of experiments using a different training set* is presented in the Appendix B.1.

## 4.1 EXPERIMENTAL SETTINGS

**Training datasets.** All models are trained on the GRAB dataset (Taheri et al., 2020). We follow the cross-object splitting strategy used in TOCH (Zhou et al., 2022) and train models on the training set. Our denoising model only requires ground-truth sequences for training. For those where the noisy counterparts are demanded, we perturb each sequence by adding Gaussian noise on the hand MANO translation, rotation, and pose parameters with standard deviations set to 0.01, 0.1, 0.5 respectively.

**Evaluation datasets.** We evaluate our model and baselines on four distinct test sets, namely GRAB test set with Gaussian noise, GRAB (Beta) test set with noise sampled from a Beta distribution ($B(8, 2)$), HOI4D dataset (Liu et al., 2022) with real noise patterns resulting from depth sensing errors and inaccurate pose estimation algorithms, and ARCTIC dataset (Fan et al., 2023) with Gaussian noise but containing challenging bimanual and dynamic interactions with changing contacts. Noisy trajectories with synthetic noise are created by adding noise sampled from corresponding distributions to the MANO parameters.

**Metrics.** We introduce two sets of evaluation metrics. The first set focuses on assessing the model's ability to recover GT trajectories from noisy inputs following previous works (Zhou et al., 2022),

including *Mean Per-Joint/Vertex Position Error (MPJPE/MPVPE)*, measuring the average distance between the denoised hand joints or vertices and the corresponding GT positions and *Contact IoU (C-IoU)* assessing the similarity between the contact map induced by denoised trajectory and the GT. The second set quantifies the quality of denoised results, including *Solid Intersection Volume (IV)* and *Penetration Depth*, measuring penetrations, *Proximity Error*, evaluating the difference of the hand-object proximity between the denoised trajectory and the GT, and *HO Motion Consistency*, assessing the hand-object motion consistency. Detailed calculations are presented in the Appendix C.2.

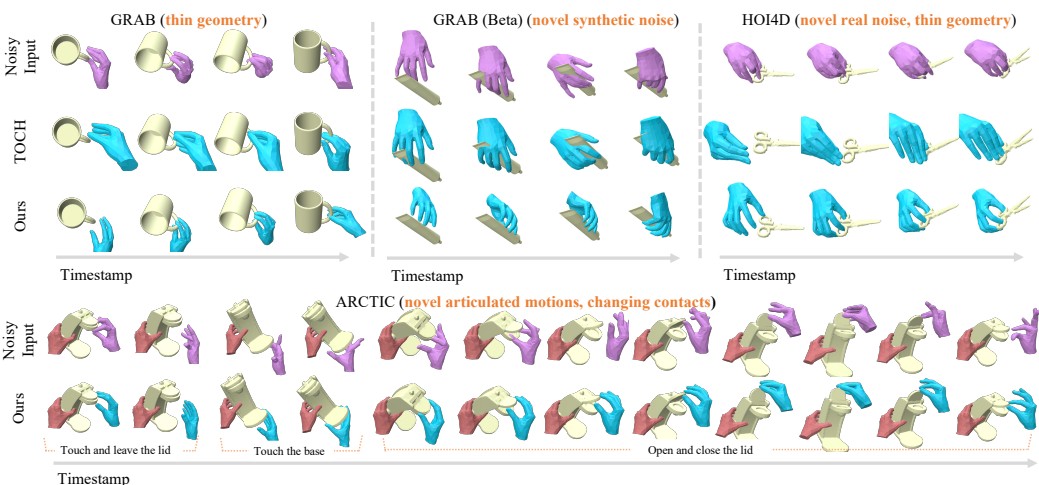

Figure 4: **Qualitative comparisons.** Please refer to **our website** and **video** for animated results.

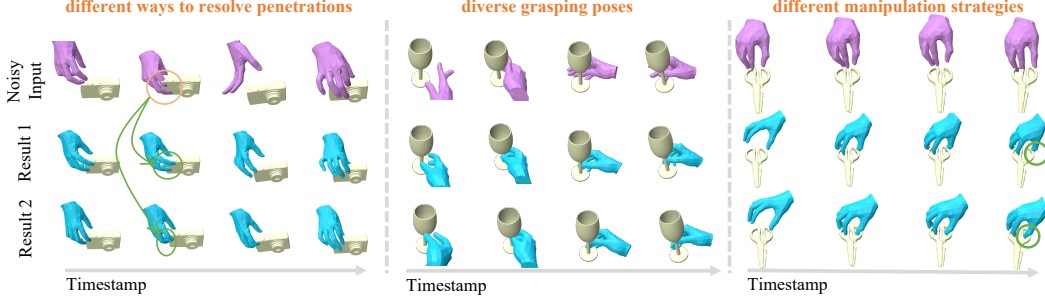

Figure 5: **Stochastic denoising** can produce diverse results with discrete modes.

**Baselines.** We compare our model with the prior art on the HOI denoising problem, TOCH (Zhou et al., 2022). A variant named "TOCH (w/ MixStyle)" is further created by combining TOCH with a general domain generalization method MixStyle (Zhou et al., 2021a). Another variant, "TOCH (w/ Aug.)", where TOCH is trained on the training sets of the GRAB and GRAB (Beta), is further introduced to enhance its robustness towards unseen noise patterns.

**Evaluation settings.** When evaluating our model, we select the trajectory that is *closest to the input noisy trajectory* from 100 randomly sampled denoised trajectories using seeds from 0 to 99. For deterministic denoising models, we report the performance on a single run. Since our model can give multiple solutions for a single input, we additionally report the performance of our model in the form of *average with standard deviations in the Appendix* on the second metric set measuring quality.

## 4.2 HOI DENOISING

We evaluated our model and compared it with previous works on four test sets: GRAB, GRAB (Beta), HOI4D, and ARCTIC. In the GRAB test set, all objects were unseen during training, resulting in a shift in the interaction distribution. In the GRAB (Beta) test set, the object shapes, interaction patterns, and noise patterns differ from those in the training set. The HOI4D dataset includes interaction sequences with novel objects and unobserved interactions, along with real noise

Table 1: **Quantitative evaluations and comparisons to baselines.** Bold red numbers for best values and *italic blue* values for the second best-performed ones. "GT" stands for "Ground-Truth".

| Dataset | Method | MPJPE ($mm, \downarrow$) | MPVPE ($mm, \downarrow$) | C-IoU ($\%, \uparrow$) | IV ($cm^3, \downarrow$) | Penetration Depth ($mm, \downarrow$) | Proximity Error ($mm, \downarrow$) | HO Motion Consistency ($mm^2, \downarrow$) |
|---|---|---|---|---|---|---|---|---|
| GRAB | GT | - | - | - | 0.50 | 1.33 | - | 0.51 |
| | Input | 23.16 | 22.78 | 1.01 | 4.48 | 5.25 | 13.29 | 881.23 |
| | TOCH | 12.38 | 12.14 | 23.31 | 2.09 | 2.17 | 3.12 | *20.37* |
| | TOCH (w/ MixStyle) | 13.36 | 13.03 | *23.70* | 2.28 | 2.62 | *3.10* | 21.29 |
| | TOCH (w/ Aug.) | *12.23* | *11.89* | 22.71 | *1.94* | *2.04* | 3.16 | 22.58 |
| | Ours | **9.28** | **9.22** | **25.27** | **1.23** | **1.74** | **2.53** | **0.57** |
| GRAB (Beta) | Input | 17.65 | 17.40 | 13.21 | 2.19 | 4.77 | 5.83 | 27.58 |
| | TOCH | 24.10 | 22.90 | 16.32 | 2.33 | 2.77 | 5.60 | 25.05 |
| | TOCH (w/ MixStyle) | 22.79 | 21.19 | 16.28 | 2.01 | 2.63 | 4.65 | 17.37 |
| | TOCH (w/ Aug.) | *11.65* | *10.47* | *24.81* | *1.52* | *1.86* | *3.07* | *13.09* |
| | Ours | **9.09** | **8.98** | **26.76** | **1.19** | **1.69** | **2.74** | **0.52** |
| HOI4D | Input | - | - | - | 2.26 | 2.47 | - | 46.45 |
| | TOCH | - | - | - | *4.09* | *4.46* | - | 35.93 |
| | TOCH (w/ MixStyle) | - | - | - | 4.31 | 4.96 | - | *25.67* |
| | TOCH (w/ Aug.) | - | - | - | 4.20 | 4.51 | - | 25.85 |
| | Ours | - | - | - | **1.99** | **2.15** | - | **9.81** |
| ARCTIC | GT | - | - | - | 0.33 | 0.92 | 0 | 0.41 |
| | Input | 25.51 | 24.84 | 1.68 | 2.28 | 4.89 | 15.21 | 931.69 |
| | TOCH | 14.34 | 14.07 | 20.32 | 1.84 | 2.01 | 4.31 | 18.50 |
| | TOCH (w/ MixStyle) | *13.82* | *13.58* | *21.70* | 1.92 | 2.13 | *4.25* | *18.02* |
| | TOCH (w/ Aug.) | 14.18 | 13.90 | 20.10 | *1.75* | *1.98* | 5.64 | 22.57 |
| | Ours | **11.57** | **11.09** | **23.49** | **1.35** | **1.93** | **2.71** | **0.92** |

caused by inaccurate sensing and vision estimations. The ARCTIC dataset contains challenging bimanual dexterous HOI sequences with dynamic contacts. Table 1 and Figure 4, 5 summarize the quantitative results and can demonstrate the superiority of our method to recover GT sequences and produce high-quality results compared to previous baseline methods. We include **more results in the Appendix B.1, our website and video** .

**Performance on challenging noisy interactions.** As shown in Figure 4, the perturbed noisy trajectories exhibit obvious problems such as unnatural hand poses, large and difficult penetrations such as penetrating the thin mug handle, and unrealistic manipulations caused by incorrect contacts and inconsistent hand-object motions. Our method can produce visually appealing interaction sequences from noisy inputs effectively. Besides, we do not have difficulty in handling difficult shapes such as the mug handle and scissor rings which are very easy to penetrate. However, TOCH cannot perform well. Its results still exhibit obvious penetrations (the last frame) and hand motions that are insufficient to manipulate the mug. Furthermore, we are not challenged by difficult and dynamic motions with changing contacts, as demonstrated by results on the ARCTIC dataset.

**Results on noisy interactions with unseen noise patterns.** In Figure 4, we demonstrate our method's robustness against new noise patterns, including previously unseen synthetic noise and novel real noise. Our approach effectively cleans such noise, producing visually appealing and motion-aware results with accurate contacts. In contrast, TOCH fails in these scenarios, as it exhibits obvious penetrations (as seen in the middle example) and results in stiff hand trajectories without proper contacts to manipulate the object (as seen in the rightmost example).

### 4.3 STOCHASTIC HOI DENOISING

Figure 5 illustrates our ability to provide multiple plausible denoised results for a single noisy input. Notably, we observe discrete manipulation modes among these results. For instance, in the leftmost example of Figure 5, our model generates different hand poses to address the unnatural phenomenon in the second frame, where two fingers penetrate through the camera. Similarly, in the rightmost example, our results offer two distinct ways to rotate the scissor for a certain angle.

### 4.4 APPLICATIONS

**Cleaning hand trajectory estimations.** As a denoising model, our approach can effectively refine hand trajectory estimations derived from image sequence observations. Figure 6 provides examples of applying our model to estimations obtained from ArcticNet-LSTM (Fan et al., 2023).

**Refining noisy retargeted hand motions.** In the right part of Figure 6, we showcase the application of our denoising model in cleaning noisy retargeted hand trajectories. Our model excels at resolving penetrations present in the sequence resulting from direct retargeting. In contrast, TOCH's result still suffers from noticeable penetrations.

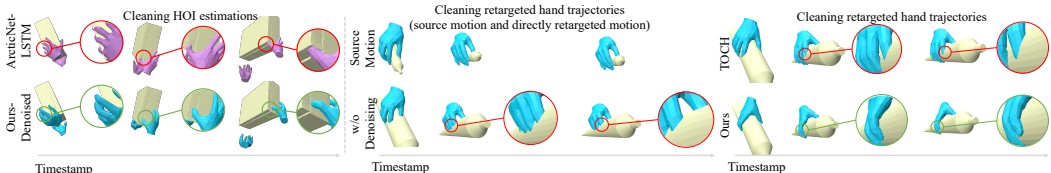

Figure 6: **Applications** on refining **noisy hand trajectories estimated from videos** (left) and cleaning **retargeted hand trajectories** (right).

## 5 ABLATION STUDY

**Generalized contact-centric parameterizations.** GeneOH leverages generalized contact points to normalize the hand-object relations. To assess the effectiveness of this design, We create an ablated model named "Ours (w/o Canon.)", which uses points sampled from the entire object surface for parameterizing. From Table 2, we can observe that our design on parameterizing around the interaction region can successfully improve the model's generalization ability towards unseen interactions.

**Denoising via diffusion.** To further investigate the impact of the "denoising via diffusion" strategy on enhancing the model's generalization ability, we ablate it by replacing the denoising model with an autoencoder structure. The results are summarized in Table 2. Besides, the comparisons between "Ours (w/o Diffusion)" and TOCH highlight the superiority of our representation GeneOH as well.

Table 2: **Ablation studies** on the HOI4D dataset.

| Method | IV ($cm^3$, ↓) | Penetration Depth ($mm$, ↓) | HO Motion Consistency ($mm^2$, ↓) |
|---|---|---|---|
| Input | 2.26 | 2.47 | 46.45 |
| Ours (w/o SpatialDiff) | 2.94 | 3.45 | 31.67 |
| Ours (w/o TemporalDiff) | **1.72** | **1.90** | 34.25 |
| Ours (w/o Diffusion) | 3.16 | 3.83 | 18.65 |
| Ours (w/o Canon.) | 2.36 | 3.57 | *13.26* |
| Ours | *1.99* | *2.15* | **9.81** |

**Hand-object spatial and temporal denoising.** We propose a progressive denoising strategy composed of three stages to clean the complex interaction noise. This multi-stage approach is crucial, as a single denoising stage would fail to produce reasonable results in the presence of complex interaction noise. To validate the effectiveness of the stage-wise denoising, we created two ablated versions: a) "Ours (w/o TemporalDiff)" by removing the temporal denoising module, and b) "Ours (w/o SpatialDiff)" by removing both the temporal and spatial denoising modules. Figure 7 and Table 2 demonstrate their effectiveness in removing unnatural hand-object penetrations and enforcing consistent hand-object motions.

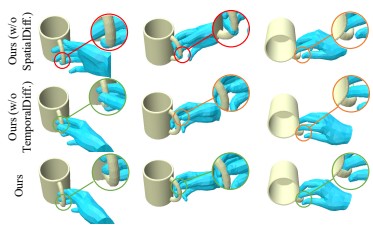

Figure 7: Effectiveness of the **SpatialDiff** and **TemporalDiff** stages.

*More quantitative and qualitative results for ablation studies* are included in the Appendix B.2.

## 6 CONCLUSION AND LIMITATIONS

In this work, we propose GeneOH Diffusion to tackle the generalizable HOI denoising problem. We resolve the challenge by 1) designing an informative HOI representation that is friendly for generalization, and 2) learning a canonical denoising model for domain-generalizable denoising. Experiments demonstrate our high denoising capability and generalization ability.

**Limitations.** The main limitation lies in the assumption of accurate object pose trajectories. It may not hold if the HOI sequences are estimated from in-the-wild videos. Refining object poses and hand poses at the same time is a valuable and practical research direction.

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

**Overview.** The **Appendix** provides a list of materials to support the main paper.

- **Additional Technical Explanations (Sec. A).** We give additional explanations to complement the main paper.

  - *The GeneOH representation (Sec. A.1).* We discuss more insights into the representation design, why GeneOH can highlight errors, and how it compares to representations designed in previous works.

  - *GeneOH Diffusion (Sec. A.2).* We talk more about the whitened noise space, the diffusion process, the multi-step stochastic denoising process, the "denoising via diffusion" strategy, and the *progressive denoising*, together with a discussion on why each denoising stage can successfully clean the input without breaking the naturalness achieved after previous stages.

  - *Fitting for a hand mesh trajectory (Sec. A.3).* We provide details of the fitting process.

- **Additional Experimental Results (Sec. B).** We include more experimental results in this section to support the effectiveness of the method, including

  - *HOI denoising results (Sec. B.1).* We include more denoising results on GRAB, GRAB (Beta), HOI4D, and the ARCTIC dataset, including *long sequences with bimanual manipulations*. Besides, we discuss the results of *another series of experiments* where the training dataset is changed to the training set of the *ARCTIC dataset*.

  - *Ablation studies (Sec. B.2).* We provide more quantitative and qualitative results of the ablation studies.

  - *Applications (Sec. B.3).* We provide more results on the applications that our model can support.

  - *Failure cases (Sec. B.4).* We discuss the limitations and failure cases of our method.

  - *Analyzing the distinction between noise in real hand-object interaction trajectories and artificial noise (Sec. B.5).* We discuss the differences between the real noise patterns and the artificial noise.

  - *User study (Sec. B.6).* We additionally include a user study to further assess the quality of our denoised results.

- **Experimental Details (Sec. C).** We illustrate details of datasets, metrics, baselines, models, the training and evaluation settings, and the running time as well as the complexity analysis.

We include a **video** and an **website** to introduce our work. The website and the video contain *animated denoised results*. We highly recommend exploring these resources for an intuitive understanding of the challenges, the effectiveness of our model, and its superiority over prior approaches.

## A ADDITIONAL TECHNICAL EXPLANATIONS

### A.1 THE GENEOH REPRESENTATION

More insights of the canonicalization design on $\mathcal{T}$ are explained as follows.

**Canonicalization design on the temporal relations $\mathcal{T}$.** The temporal relations $\mathcal{T}$ leverages hand-object relative velocity $\mathbf{v}_k^{\mathbf{ho}}$ at each frame $k$ of each hand-object point pair $(\mathbf{h}, \mathbf{o})$ to represent the motion relations. We further canonicalize the relative velocity via object normals by decomposing $\mathbf{v}_k^{\mathbf{ho}}$ into two statistics: $\mathbf{v}_{k,\perp}^{\mathbf{ho}}$, vertical to the object tangent plane and parallel to the object point normal, and $\mathbf{v}_{k,\parallel}^{\mathbf{ho}}$, lying in the object's tangent plane. This decomposition enables the model to learn different denoising strategies for the two types of relative velocities and enhance cross-interaction generalization by factoring out object poses. However, relying solely on relative velocities is insufficient to reveal motion noise in hand-object interactions. The same relative velocity parallel to the normal direction can correspond to a clean state when the hand is far from the object, but a noisy state when they are in contact. To address this, we further encode the distance between each hand-object pair and their relative velocities into two statistics, $\{(e_{k,\perp}^{\mathbf{ho}}, e_{k,\parallel}^{\mathbf{ho}})\}$, using the following

formulation:

$$e_{k,\perp}^{\mathbf{ho}} = e^{-k \cdot d_k^{\mathbf{ho}}} k_b \|\mathbf{v}_{k,\perp}^{\mathbf{ho}}\|_2 \tag{2}$$

$$e_{k,\|}^{\mathbf{ho}} = e^{-k \cdot d_k^{\mathbf{ho}}} k_a \|\mathbf{v}_{k,\|}^{\mathbf{ho}}\|_2. \tag{3}$$

Here, $k$, $k_a$, and $k_b$ are positive hyperparameters, and the term $e^{-k \cdot d_k^{\mathbf{ho}}}$ is inversely related to the distance between the hand and object points. This formula allows the statistics for very close hand-object point pairs to be emphasized in the representation.

**Why GeneOH can highlight errors?** For spatial errors, we mainly consider the geometric penetrations between the hand and the object. The hand-object spatial relations $\mathcal{S}$ in GeneOH reveal penetrations by parameterizing the object normals and hand-object relative offsets. In more detail, for each hand-object point pair $(\mathbf{h}_k, \mathbf{o}_k)$, the dot product between the object normal $\mathbf{n}_k$ of $\mathbf{o}_k$ and the relative offset $\mathbf{h}_k - \mathbf{o}_k$ indicates the signed distance between the object point and the hand point. For each hand point $\mathbf{h}_k$, its signed distance to the object mesh can be revealed by jointly considering its signed distance to all generalized contact points. Since the hand point $\mathbf{h}_k$ penetrates the object *if and only if its signed distance to the object mesh is negative*, the spatial relation parameterization $\mathcal{S}$ can indicate the penetration phenomena.

For temporal errors, we mainly consider inconsistent hand-object motions. There is no unified definition or statement regarding what consistent hand-object motions indicate. Intuitively, the hand should be able to manipulate the object, where sufficient contact and consistent motions between very close hand-object point pairs are demanded. For very close hand-object pairs, the sliding motion on the object surface is permitted but the vertical penetration moving tendency is not allowed. The above expectations and unnatural situations can be revealed from simple statistics like hand-object relative velocities and hand-object distances. The distance can tell whether they are close to each other. The relative velocity can reveal their moving discrepancy. The decomposed relative velocity lying in the tangent plane and vertical to the tangent plane indicate the surface sliding tendency and the penetrating tendency respectively. The distance-related weight term $e^{-k_f d_k^{\mathbf{ho}}}$ can emphasize hand-object pairs that are very close to each other. Therefore, the temporal relations representation $\mathcal{T}$ leveraged in GeneOH can successfully indicate the temporal naturalness and incorrect phenomena. Thus learning the distribution of $\mathcal{T}$ can teach the model what is temporal naturalness and how to clean the noisy representation.

**The GeneOH representation** can be applied to parameterize interaction sequences involving rigid or articulated objects. It carefully integrates both the hand motions and hand-object spatial and temporal relations — more expressive and comprehensive compared to designs in previous works (Zhou et al., 2022; 2021b; Zhang et al., 2021). Besides, it can highlight spatial and temporal interaction errors. The above advantages make GeneOH well-suited for the HOI denoising task.

Inspecting back to previous works, TOCH (Zhou et al., 2022) does not explicitly encode hand-centric poses or hand-object temporal relations and only grounds the hands onto the object without careful consideration of cross-interaction alignment. ManipNet (Zhang et al., 2021) takes hand-object distances to represent their relations. But this is not enough to reveal their spatial relations. Canonicalizations are also not carefully considered in this work.

## A.2 GENEOH DIFFUSION

We give a more detailed explanation of the three denoising stages in the following text.

**The whitened noise space.** This space is constructed by diffusing the training data towards a random Gaussian noise. A diffusion timestep $1 \leq t_{\text{diff}} \leq T$ where $T$ is the maximum timestep controls to what extent the input is diffused to pure Gaussian noise. It is the space modeled by the diffusion models during training. The diffusion function we adopt in this work is also exactly the same as the forward diffusion process of diffusion models. To be more specifically, given a data point $x$, the $t_{\text{diff}}$ diffusion would transform to $x_t$ by a linear combination of $x$ and a random Gaussian noise $\mathbf{n}$ with the same size to $x$ via the following equation:

$$x_t = \sqrt{\bar{\alpha}_t} x + \sqrt{1 - \bar{\alpha}_t} \mathbf{n}, \tag{4}$$

where $\alpha_t = 1 - \beta_t$, $\bar{\alpha}_t = \Pi_{s=1}^t \alpha_s$, $\{\beta_t\}$ is the forward process variances. The distribution of $x_t$ is a normal distribution: $x_t \sim \mathcal{N}(\sqrt{\bar{\alpha}_t} x, (1 - \sqrt{\bar{\alpha}_t})\mathbf{I})$.

Intuitively, the noise space contains all possible $x_t$ across all possible timestep $1 \leq t \leq T$. $x_t$ with smaller $t$ will be more similar to $x$. In practice, $T$ is set to 1000, $\beta_1 = 0.001, \beta_T = 0.02$ with a linear interpolation between them to create the variance sequence $\{\beta_t\}$. During training $t$ is uniformly sampled from the $\{t | t \in \mathbb{Z}, 1 \leq t \leq T\}$.

**Training of the denoising model.** Denote the multi-step stochastic denoising model leveraged in our method as $\text{denoise}(\cdot, t)$ which takes the noisy sample with the noise scale $t$ as input and denoise it back to the noise sample with noise scale $t - 1$. When $t = 1$, the denoised result lies in the clean data manifold depicted by the model and is taken as the final denoised result. The $\text{denoise}(\cdot, t)$ leverages a score function $\epsilon_\theta(\cdot, t)$ to predict the noise component of the input noise sample $\tilde{x}_t$. The score function $\epsilon_\theta(\cdot, t)$ contains optimizable network weights $\theta$ and is what we need to learn during training. $\epsilon_\theta(\cdot, t)$ only predicts the noise component $\hat{\mathbf{n}}$. After that, a posterior sampling process is leveraged to sample $\tilde{x}_{t-1}$ based on $\tilde{x}_t$ and the predicted $\hat{\mathbf{n}}$ via the following equation:

$$\tilde{x}_{t-1} = \frac{1}{\sqrt{\alpha_t}}(\tilde{x}_t - \frac{1 - \alpha_t}{\sqrt{1 - \bar{\alpha}_t}}\epsilon_\theta(\tilde{x}_t, t)) + \sigma_t \mathbf{z}, \tag{5}$$

where $\mathbf{z} \in \mathcal{N}(\mathbf{0}, \mathbf{I})$, $\sigma_t^2 = \beta_t$. Therefore, the denoising model is a multi-step stochastic denoiser since at each step it only identifies the mean of the posterior distribution and the denoised result needs to be sampled from the distribution with the predicted mean and the pre-defined variance.

**The "denoising via diffusion" strategy.** The input trajectory with noise $\hat{x}$ is diffused to $\tilde{x}$ via Gaussian noise with the diffusion timestep $t_{\text{diff}}$ using the following equation:

$$\tilde{x} = \tilde{x}_{t_{\text{diff}}} = \hat{x}_{t_{\text{diff}}} = \sqrt{\bar{\alpha}_{t_{\text{diff}}}}x + \sqrt{1 - \bar{\alpha}_{t_{\text{diff}}}}\mathbf{n}, \tag{6}$$

where $\mathbf{n} \in \mathcal{N}(\mathbf{0}, \mathbf{I})$ is a random Gaussian noise.

Given the noisy sample $\tilde{x}_t$ with noise scale $t$, the denoising model $\text{denoise}(\cdot, t)$ predicts its noise component via the score function :

$$\hat{\mathbf{n}} = \epsilon_\theta(\tilde{x}_t, t). \tag{7}$$

Then $\tilde{x}_t$ is denoised to $\tilde{x}_{t-1}$ with noise scale $t - 1$ by sampling from the posterior distribution following Eq. 5 with the predicted mean $\tilde{x}_t - \frac{1 - \alpha_t}{\sqrt{1 - \bar{\alpha}_t}}\hat{\mathbf{n}}$ and the pre-defined variance $\sigma_t^2$.

**MotionDiff: canonicalized hand trajectory denoising.** This denoising stage removes noise from the canonicalized hand trajectory $\hat{\bar{\mathcal{J}}}^{\text{input}}$ of the input noisy interaction sequence by applying the diffusion model for one stage-denoised $\bar{\mathcal{J}}^{\text{stage}_1}$, following the "denoising via diffusion" strategy. To do this, the noisy representation $\hat{\bar{\mathcal{J}}}^{\text{input}}$ is diffused by adding noise for $t_m$ steps, followed by denoising for $t_m$ steps using the diffusion model. The resulting one stage-denoised hand trajectory $\mathcal{J}^{\text{stage}_1}$ in the world coordinate space is obtained by de-canonicalizing the denoised canonicalized hand trajectory $\bar{\mathcal{J}}^{\text{stage}_1}$ using the pose of the generalized contact points $(\mathbf{R}_k, \mathbf{t}_k)$. $\text{GeneOH}^{\text{input}}$ can also be updated accordingly into $\text{GeneOH}^{\text{stage}_1} = \{\bar{\mathcal{J}}^{\text{stage}_1}, \hat{\mathcal{S}}^{\text{stage}_1}, \hat{\mathcal{T}}^{\text{stage}_1}\}$.

**SpatialDiff: hand-object spatial denoising.** The hand-object spatial denoising module operates on the noisy hand-object spatial relations $\hat{\mathcal{S}}^{\text{stage}_1}$ of the one stage-denoised interaction sequence output by the previous MotionDiff stage. The representation $\hat{\mathcal{S}}^{\text{stage}_1}$ is diffused by adding noise for $t_s$ diffusion steps, followed by another $t_s$ step of denoising. Once we obtain the denoised representation $\mathcal{S}^{\text{stage}_2}$ which includes the hand-object relative offsets $\{(\mathbf{h}_k - \mathbf{o}_k)\}$ centered at each generalized contact point $\mathbf{o}_k$, we adopt a simple approach to convert it into a two stages-denoised hand sequence. Specifically, we average the denoised hand offsets from each object point as follows:

$$\mathcal{J} = \text{Average}\{(\mathbf{h}_k - \mathbf{o}_k) + \mathbf{o}_k | \mathbf{o}_k \in \mathbf{P}_k\}. \tag{8}$$

Following this, $\text{GeneOH}^{\text{stage}_1}$ will be updated to $\text{GeneOH}^{\text{stage}_2} = \{\bar{\mathcal{J}}^{\text{stage}_2}, \mathcal{S}^{\text{stage}_2}, \hat{\mathcal{T}}^{\text{stage}_2}\}$.

**TemporalDiff: hand-object temporal denoising.** We proceed to clean the noisy hand-object temporal relations $\hat{\mathcal{T}}^{\text{stage}_2}$ of the two stages-denoised sequence. The "denoising via diffusion" procedure is applied to the temporal relations to achieve this. We then add an additional optimization to distill the information contained in the denoised temporal representation to the three stages-denoised trajectory. The objective is formulated as:

$$\underset{\mathcal{J}^{\text{stage}_2}}{\text{minimize}}\|f_{(\mathcal{J}, \mathcal{P}) \rightarrow \mathcal{T}}(\mathcal{J}^{\text{stage}_2}, \mathcal{P}) - \mathcal{T}^{\text{stage}_3}\|, \tag{9}$$

where $\mathcal{P}$ is the sequence of generalized contact points, $f_{(\mathcal{J},\mathcal{P})\to\mathcal{T}}(\cdot,\cdot)$ converts the hand trajectory to the corresponding temporal relations. The distance is calculated on hand-object distances, *i.e.,* $\{d_k^{\mathbf{ho}}\}$, relative velocity $\{\mathbf{v}_k^{\mathbf{ho}}\}$ and two relative velocity-related statistics ( $\{e_{k,\|}^{\mathbf{ho}}, e_{k,\perp}^{\mathbf{ho}}\}$ ). We employ an Adam optimizer to find the optimal hand trajectory. The optimized trajectory is taken as the final denoised trajectory $\mathcal{J}$.

**Stage-wise denoising strategy.** Let $\mathcal{I} = \{(\mathcal{H}_k, \mathbf{O}_k)\}_{k=1}^K \in \mathcal{M}$ denote an interaction sequence, where $\mathcal{M}$ is the manifold contains all interaction sequences. Let $\mathcal{M}_{\bar{\mathcal{J}}}$, $\mathcal{M}_{\mathcal{S}}$, and $\mathcal{M}_{\mathcal{T}}$ represent the manifolds depicted by the three denoising stages respectively. Let $\mathcal{I}_{\bar{\mathcal{J}}}$, $\mathcal{I}_{\mathcal{S}}$, and $\mathcal{I}_{\mathcal{T}}$ represent one stage-denoised trajectory, two stages-denoised trajectory, and the three stages-denoised trajectory respectively. Further, let $\mathcal{R}_{\bar{\mathcal{J}}}^c$, $\mathcal{R}_{\mathcal{S}}^c$, and $\mathcal{R}_{\mathcal{T}}^c$ denote the set of all natural canonicalized hand trajectories, natural hand-object spatial relations, and correct hand-object temporal relations respectively. Let $\mathcal{R}_{\bar{\mathcal{J}}}$, $\mathcal{R}_{\mathcal{S}}$, and $\mathcal{R}_{\mathcal{T}}$ denote the set of all canonicalized hand trajectories, hand-object spatial relations, and hand-object temporal relations respectively. Denote the function that transforms the interaction trajectory $\mathcal{I}$ to the canonicalized hand trajectory as $f_{\mathcal{I}\to\bar{\mathcal{J}}}$, the function that converts $\mathcal{I}$ to the hand-object spatial relations as $f_{\mathcal{I}\to\mathcal{S}}$, and the function that transforms $\mathcal{I}$ to the hand-object temporal relations as $f_{\mathcal{I}\to\mathcal{T}}$.

For all interaction trajectories considered in the work, we make the following assumption:

**Assumption** *For any trajectory $\mathcal{I}$ with the first frame free of spatial noise, we can find a natural trajectory $\mathcal{I}'$ with the same first frame, that is $\mathcal{I}'[1] = \mathcal{I}[1]$.*

The three fully-trained denoising models for $\bar{\mathcal{J}}$, $\mathcal{S}$, and $\mathcal{T}$ should be able to map the corresponding input representation to the set of $\mathcal{R}_{\bar{\mathcal{J}}}^c$, $\mathcal{R}_{\mathcal{S}}^c$, and $\mathcal{R}_{\mathcal{T}}^c$ respectively. Then the relations between the interaction manifolds depicted by the three denoising stages and the natural data prior modeled by the three denoising models have the following relations:

- $\mathcal{I} \in \mathcal{M}_{\bar{\mathcal{J}}}$ if and only if $f_{\mathcal{I}\to\bar{\mathcal{J}}}(\mathcal{I}) \in \mathcal{R}_{\bar{\mathcal{J}}}^c$;
- $\mathcal{I} \in \mathcal{M}_{\mathcal{S}}$ if and only if $f_{\mathcal{I}\to\mathcal{S}}(\mathcal{I}) \in \mathcal{R}_{\mathcal{S}}^c$;
- $\mathcal{I} \in \mathcal{M}_{\mathcal{T}}$ if and only if $f_{\mathcal{I}\to\mathcal{T}}(\mathcal{I}) \in \mathcal{R}_{\mathcal{T}}^c$ and the first frame $\mathcal{I}[1]$ is free of spatial noise.

Based on the relations between $\bar{\mathcal{J}}$, $\mathcal{S}$, and $\mathcal{T}$, we can make the following claim:

**Claim 1** *There existing functions $f_{\mathcal{S}\to\bar{\mathcal{J}}} : \mathcal{R}_{\mathcal{S}} \to \mathcal{R}_{\bar{\mathcal{J}}}$ and $f_{\mathcal{T}\to\mathcal{S}} : (\mathcal{R}_{\mathcal{T}}, \mathcal{I}_{\mathcal{S}[1]} \to \mathcal{R}_{\mathcal{S}})$, so that for any interaction $\mathcal{I}$ with corresponding GeneOH representations $GeneOH(\mathcal{I}) = \{\bar{\mathcal{J}}, \mathcal{S}, \mathcal{T}\}$, we have: $\bar{\mathcal{J}} = f_{\mathcal{S}\to\bar{\mathcal{J}}}(\mathcal{S})$ and $\mathcal{S} = f_{\mathcal{T}\to\mathcal{S}}(\mathcal{T}, \mathcal{I}[1])$.*

*Proof. The canonicalized hand keypoints at each frame $k$, i.e., $\bar{\mathbf{J}}_k$ is composed of each canonicalized hand keypoint $\bar{\mathbf{J}}_k = \{(\mathbf{h}_k - \mathbf{t}_k)\mathbf{R}_k^T | \mathbf{h}_k \in \mathbf{J}_k\}$, which can be derived from the canonicalized hand-object spatial relation at the frame $k$. Specifically, for each $\mathbf{o}_k \in \mathbf{P_k}$, we have $(\mathbf{h}_k - \mathbf{t}_k)\mathbf{R}_k^T = (\mathbf{h}_k - \mathbf{o}_k)\mathbf{R}_k^T + (\mathbf{o}_k - \mathbf{t}_k)\mathbf{R}_k^T$. The unique canonicalized hand trajectory at the frame $k$ can be decided from the trajectory converted from each object point $\mathbf{o}_k \in \mathbf{P}_k$. Depending on the conversion function from such multiple hypotheses of the canonicalized hand trajectory resulting from different $\mathbf{o}_k$, there exists a function $f_{\mathcal{S}\to\bar{\mathcal{J}}} : \mathcal{S} \to \mathcal{R}_{\bar{\mathcal{J}}}$ that transforms the hand-object spatial relations $\mathcal{S}$ to the canoncialized hand trajectory $\bar{\mathcal{J}}$.*

*Similarly, given the hand-object temporal relations at the frame $k(1 \le k \le K - 1)$ of the object point $\mathbf{o}_k \in \mathbf{P}_k$ and the natural hand keypoints at the starting frame $1$, i.e., $\mathbf{J}_1$, the relative velocity for each hand-object pair $(\mathbf{h}_k, \mathbf{o}_k)$ can be derived from the decoded hand-object relative velocity $\mathbf{v}_k^{\mathbf{ho}}$, two velocity-related statistics $(e_{k,\perp}^{\mathbf{ho}}, e_{k,\|}^{\mathbf{ho}})$, the hand-object distance $d_k^{\mathbf{ho}}$. Given the hand-object relative positions $\{(\mathbf{h}_1 - \mathbf{o}_1) | \mathbf{h}_1 \in \mathbf{J}_1\}$, the hand-object relative positions at each following frame $k + 1(1 \le k \le K - 1)$ can be derived iteratively via the hand-object relative velocity $\{\mathbf{v}_k^{\mathbf{ho}} | \mathbf{o}_k \in \mathbf{P}_k\}$: $\mathbf{h}_{k+1} - \mathbf{o}_{k+1} = (\mathbf{h}_k - \mathbf{o}_k) + \Delta t \mathbf{v}_k^{\mathbf{ho}}$. Therefore, there existing a function $f_{\mathcal{T}\to\mathcal{S}} : \mathcal{R}_{\mathcal{T}} \to \mathcal{R}_{\mathcal{S}}$ that can convert the temporal relations $\mathcal{T}$ to the hand-object spatial relations $\mathcal{S}$.* ∎

Based on this property, we can make the following claim regarding the relations between the three gradually constructed manifolds:

**Claim 2** *Assume the first frame of the two stages-denoised trajectory $\mathcal{I}_{\mathcal{S}}[1]$ is free of spatial noise, which **almost always holds true**, we have $\mathcal{M}_{\mathcal{T}} \subseteq \mathcal{M}_{\mathcal{S}} \subseteq \mathcal{M}_{\bar{\mathcal{J}}}$.*

**Proof.** *For $\mathcal{I} \in \mathcal{M}_{\mathcal{S}}$ with the GeneOH representation $\{\bar{\mathcal{J}}, \mathcal{S}, \mathcal{T}\}$, assume $\mathcal{I} \notin \mathcal{M}_{\bar{\mathcal{J}}}$.*

- *From $\mathcal{I} \in \mathcal{M}_{\mathcal{S}}$, we have $\mathcal{S} \in \mathcal{R}_{\mathcal{S}}^c$;*

- *Based on the definition of $\mathcal{R}_{\mathcal{S}}^c$, the set of spatial relations derived from all natural interactions, there exists a natural interaction $\mathcal{I}'$ so that $f_{\mathcal{I} \to \mathcal{S}}(\mathcal{I}') = \mathcal{S}$;*

- *Since $\mathcal{I}'$ is a natural interaction, we have $\bar{\mathcal{J}}' = f_{\mathcal{I} \to \bar{\mathcal{J}}}(\mathcal{I}') \in \mathcal{R}_{\bar{\mathcal{J}}}^c$;*

- *Since $\bar{\mathcal{J}} = f_{\mathcal{S} \to \bar{\mathcal{J}}}(\mathcal{S}) = \bar{\mathcal{J}}'$, we have $\bar{\mathcal{J}} \in \mathcal{R}_{\bar{\mathcal{J}}}^c$;*

- *Based on the assumed fully-trained denoising model, we have $\mathcal{I} \in \mathcal{M}_{\bar{\mathcal{J}}}$.*

*The conclusion contradicts with the assumption $\mathcal{I} \notin \mathcal{M}_{\bar{\mathcal{J}}}$. Thus $\mathcal{M}_{\mathcal{S}} \subseteq \mathcal{M}_{\bar{\mathcal{J}}}$ holds true.*

*For a $\mathcal{I} \in \mathcal{M}_{\mathcal{T}}$ with the GeneOH representation $\{\bar{\mathcal{J}}, \mathcal{S}, \mathcal{T}\}$ whose first frame is free of spatial noise, assume $\mathcal{I} \notin \mathcal{M}_{\mathcal{S}}$.*

- *From $\mathcal{I} \in \mathcal{M}_{\mathcal{T}}$, we have that $\mathcal{T} \in \mathcal{R}_{\mathcal{T}}^c$;*

- *Based on the definition of $\mathcal{R}_{\mathcal{T}}^c$, the set of temporal relations derived from all natural interactions, and the Assumption 1, there existing a natural interaction $\mathcal{I}'$, with the first frame same to $\mathcal{I}$, so that $f_{\mathcal{I} \to \mathcal{T}}(\mathcal{I}') = \mathcal{T}$;*

- *Since $\mathcal{I}'$ is a natural interaction, we have $\mathcal{S}' = f_{\mathcal{I} \to \mathcal{S}}(\mathcal{I}') \in \mathcal{R}_{\mathcal{S}}^c$;*

- *Since $\mathcal{S} = f_{\mathcal{T} \to \mathcal{S}}(\mathcal{T}, \mathcal{I}[1]) = f_{\mathcal{T} \to \mathcal{S}}(\mathcal{T}, \mathcal{I}'[1]) = \mathcal{S}'$, we have $\mathcal{S} \in \mathcal{R}_{\mathcal{S}}^c$;*

- *Based on the assumed fully-trained denoising model, we have $\mathcal{I} \in \mathcal{M}_{\mathcal{S}}$.*

*The conclusion contradicts with the assumption $\mathcal{I} \notin \mathcal{M}_{\mathcal{S}}$. Thus $\mathcal{M}_{\mathcal{T}} \subseteq \mathcal{M}_{\mathcal{S}}$ holds true.* ∎

The stage-wise GeneOH Diffusion functions as the following steps to clean the input interaction $\mathcal{I} \in \mathcal{M}$:

- Given the input interaction $\mathcal{I}$, the denoising model for $\bar{\mathcal{J}}$ maps $\bar{\mathcal{J}}$ to another $\bar{\mathcal{J}}^1 \in \mathcal{R}_{\bar{\mathcal{J}}}^c$. There existing an interaction $\mathcal{I}^1$ s.t. $\bar{\mathcal{J}}^1 = f_{\mathcal{I} \to \bar{\mathcal{J}}}(\mathcal{I}^1)$, which is also exactly the same as the trajectory derived from $\bar{\mathcal{J}}^1$ and the object trajectory $\{\mathbf{O}_k\}_{k=1}^K$. Therefore, after the first denoising stage, we have $\mathcal{I}^1 \in \mathcal{M}_{\bar{\mathcal{J}}}$.

- Given $\mathcal{S}^1$, the denoising model for $\mathcal{S}$ maps $\mathcal{S}^1$ to $\mathcal{S}^2 \in \mathcal{R}_{\mathcal{S}}^c$. There existing an interaction $\mathcal{I}^2$ s.t. $\mathcal{S}^2 = f_{\mathcal{I} \to \mathcal{S}}(\mathcal{I}^2)$. Therefore, after the second denoising stage, we have $\mathcal{I}^2 \in \mathcal{M}_{\mathcal{S}}$.

- After that, the denoising model for $\mathcal{T}$ maps $\mathcal{T}^2 = f_{\mathcal{I} \to \mathcal{T}}(\mathcal{I}^2)$ to $\mathcal{T}^3 \in \mathcal{R}_{\mathcal{T}}^c$. After that, the interaction $\mathcal{I}^3$ constructed as the following steps:
    - Construct $\mathcal{S}^3$ from $\mathcal{T}^3$ and $\mathcal{I}^2[1]$ via $\mathcal{S}^3 = f_{\mathcal{T} \to \mathcal{S}}(\mathcal{T}^3, \mathcal{I}^2[1])$;
    - Construct $\bar{\mathcal{J}}^3$ from $\mathcal{S}^3$ via $\bar{\mathcal{J}}^3 = f_{\mathcal{S} \to \bar{\mathcal{J}}}(\mathcal{S}^3)$;
    - Construct $\mathcal{I}^3$ from $\bar{\mathcal{J}}^3$ and the object trajectory $\{\mathbf{O}_k\}_{k=1}^K$.
  Since $\mathcal{T}^3 = f_{\mathcal{I} \to \mathcal{T}}(\mathcal{I}^3) \in \mathcal{R}_{\mathcal{T}}^c$ and $\mathcal{I}^3[1] = \mathcal{I}^2[1]$ is free of spatial noise, we have $\mathcal{I}^3 \in \mathcal{M}_{\mathcal{T}}$.

Therefore, the three denoising stages gradually map the input noisy interaction to a progressively smaller manifold contained in the previous large manifold. Formally we have

$$\text{GeneOH Diffusion}(\cdot) : \mathcal{M} \to \mathcal{M}_{\bar{\mathcal{J}}} \to \mathcal{M}_{\mathcal{S}} \to \mathcal{M}_{\mathcal{T}}. \tag{10}$$

### A.3 FITTING FOR AN HOI TRAJECTORY

Once the interaction sequence has been denoised, we proceed to fit a sequence of hand MANO (Romero et al., 2022) parameters to obtain final hand meshes. The objective is optimizing a series of MANO parameters $\{\mathbf{r}_k, \mathbf{t}_k, \beta_k, \theta_k\}_{k=1}^K$ so that they fit the denoised trajectory $\mathcal{J}$ well. Notice the hand trajectory $\mathcal{J}$ consists of a sequence of hand keypoints so we also need to derive

Table 3: **HOI4D Dataset.** Per-category statistics of the HOI4D dataset used in our experiments, including number of sequences and the index of the start frame.

|  | Laptop | Pliers | Scissors | Bottle | Bowl | Chair | Mug | ToyCar | Kettle |
|---|---|---|---|---|---|---|---|---|---|
| #Seq. | 155 | 187 | 93 | 214 | 217 | 167 | 249 | 257 | 58 |
| Starting Frame | 120 | 150 | 50 | 0 | 0 | 0 | 0 | 0 | 0 |

keypoints from MANO parameters to allow the above optimization. Luckily this process is differentiable and we use $\mathcal{J}^{recon}(\{\mathbf{r}_k, \mathbf{t}_k, \beta_k, \theta_k\}_{k=1}^K)$ to denote it. We can therefore optimize the following reconstruction loss for the MANO hands

$$\mathcal{L}_{recon} = \|\mathcal{J} - \mathcal{J}^{recon}(\{\mathbf{r}_k, \mathbf{t}_k, \beta_k, \theta_k\}_{k=1}^K)\|, \tag{11}$$

where the distance function is a simple mean squared error (MSE) between the hand keypoints at each frame. To regularize the hand parameters $\{\beta_k, \theta_k\}$ and enforce temporal smoothness, we introduce an additional regularization loss defined as

$$\mathcal{L}_{reg} = \frac{1}{K} \sum_{k=1}^{K} (\|\beta_k\|_2 + \|\theta_k\|_2) + \frac{1}{K-1} \sum_{k=1}^{K-1} \|\theta_{k+1} - \theta_k\|_2. \tag{12}$$

The overall optimization objective is formalized as

$$\text{minimize}_{\{\mathbf{r}_k, \mathbf{t}_k, \beta_k, \theta_k\}_{k=1}^K} (\mathcal{L}_{recon} + \mathcal{L}_{reg}), \tag{13}$$

and we employ an Adam optimizer to solve the fitting problem.

# B  ADDITIONAL EXPERIMENTAL RESULTS

We present additional experimental results to further support the denoising effectiveness and the strong generalization ability of our method.

## B.1  HOI DENOISING

For the first set of evaluation metrics, Table 4 presents more evaluations on our method, ablated versions, and the comparisons to the baseline models than the table in the main text. For the second set of evaluation metrics, Table 5 summarizes more results and comparisons. For stochastic denoising methods, including "Ours", "Ours w/o Canon.", "Ours w/o SpatialDiff", and "Ours w/o TemporalDiff", we report the mean and the standard deviation of results obtained from three independent runs with the random seed set to 11, 22, and 77 respectively. Such statistics offer a more comprehensive view of the average results quality produced by those methods. Please notice that the evaluation method is different from the one present in the main text, where the result closest to the input trajectory among 100 independent runs is chosen to report evaluation metrics.

**More results on the GRAB test set – novel interactions with new objects.** Figure 8 shows qualitative evaluations on the GRAB test set to compare the generalization ability of different denoising models towards novel interactions with unseen objects.

**More results on the GRAB (Beta) test set – novel interactions with new objects and unseen synthetic noise patterns.** Figure 9 shows qualitative evaluations on the GRAB (Beta) test set to compare the generalization ability of different denoising models towards unseen objects, unobserved interactions, and novel synthetic noise.

**More results on the HOI4D dataset – novel interactions with new objects and unseen real noise patterns.** Figure 10 shows qualitative evaluations on the HOI4D test set to compare the generalization ability of different denoising models towards unseen objects, unobserved interactions, and novel real noise.

**Wired hand trajectory produced by TOCH (w/ MixStyle).** Through our experiments with TOCH on interaction sequences with new noise patterns unseen during training, we frequently observe strange hand trajectories from its results with stiff hand poses, unsmooth trajectories, and large

Table 4: **Quantitative evaluations and comparisons.** Performance comparisons of our method, baselines, and ablated versions on different test sets using the *first set of evaluation metrics*. **Bold red** numbers for best values and *italic blue* values for the second best-performed ones.

| Dataset | Method | MPJPE $(mm, \downarrow)$ | MPVPE $(mm, \downarrow)$ | C-IoU $(\%, \uparrow)$ |
|---|---|---|---|---|
| GRAB | Input | 23.16 | 22.78 | 1.01 |
| | TOCH | 12.38 | 12.14 | 23.31 |
| | TOCH (w/ MixStyle) | 13.36 | 13.03 | 23.70 |
| | TOCH (w/ Aug.) | 12.23 | 11.89 | 22.71 |
| | Ours (w/o SpatialDiff) | **7.83** | **7.67** | 26.09 |
| | Ours (w/o TemporalDiff) | *8.27* | *8.13* | **26.55** |
| | Ours (w/o Diffusion) | 8.52 | 8.38 | *26.44* |
| | Ours (w/o Canon.) | 10.15 | 10.07 | 24.92 |
| | Ours | 9.28 | 9.22 | 25.27 |
| GRAB (Beta) | Input | 17.65 | 17.40 | 13.21 |
| | TOCH | 24.10 | 22.90 | 16.32 |
| | TOCH (w/ MixStyle) | 22.79 | 21.19 | 16.28 |
| | TOCH (w/ Aug.) | 11.65 | *10.47* | *24.81* |
| | Ours (w/o Diffusion) | 12.16 | 11.75 | 22.96 |
| | Ours (w/o Canon.) | *10.89* | 10.61 | 24.68 |
| | Ours | **9.09** | **8.98** | **26.76** |
| ARCTIC | Input | 25.51 | 24.84 | 1.68 |
| | TOCH | 14.34 | 14.07 | 20.32 |
| | TOCH (w/ MixStyle) | *13.82* | *13.58* | *21.70* |
| | TOCH (w/ Aug.) | 14.18 | 13.90 | 20.10 |
| | Ours | **11.57** | **11.09** | **23.49** |

penetrations as shown in Figure 9 and 4. Such phenomena cannot be mitigated by augmenting it with general domain generalization techniques. Figure 12 demonstrates that the improved version, TOCH with MixStyle, also yields similar unnatural results. This suggests that the novel noise distribution presents a challenging obstacle for the denoising model to generalize to new noisy interaction sequences. In contrast, our method does not have such difficulty in handling the shifted noise distribution.

**Results of TOCH and TOCH (w/ Aug.) on the HOI4D dataset.** Figure 11 compares the results of TOCH (w/ Aug.) with our method. In the example of opening a scissor, TOCH produces very strange "flying hands" trajectories, for which please refer to our video for an intuitive understanding. Though the results produced by the improved version do not exhibit the "flying hands" artifacts, it is still very strange, stiff, suffering from very unnatural hand poses, and cannot perform correct manipulations. The results of TOCH and TOCH (w/ Aug.) on the ToyCar example are very similar since our experiments indeed get very similar results from such two models in this case. They both are troubled by strange hand shapes and very unnatural trajectories.

**More results on the ARCTIC dataset – novel interactions with new objects involving dynamic object motions and changing contacts.** Figure 13 shows qualitative evaluations on the ARCTIC test set to test the ability of our denoising model to clean noisy and dynamic interactions with changing contacts.

Besides, we include samples of our results on longer sequences with bimanual manipulations in Figure 14.

**Multi-state denoising v.s. one-stage denoising.** We leverage a multi-stage denoising strategy in this work to tackle the challenge posed by complex interaction noise. In Section A.2, we demonstrate the stage-wise denoising strategy gradually projects the input trajectory from the manifold containing unnatural interactions, to the manifold of trajectories with natural hand motions, to the manifold

Table 5: **Quantitative evaluations and comparisons.** Performance comparisons of our method, baselines, and ablated versions on different test sets using the *second set of evaluation metrics*. **Bold red** numbers for best values and *italic blue* values for the second best-performed ones. "GT" stands for "Ground-Truth".

| Dataset | Method | IV $(cm^3, \downarrow)$ | Penetration Depth $(mm, \downarrow)$ | Proximity Error $(mm, \downarrow)$ | HO Motion Consistency $(mm^2, \downarrow)$ |
|---|---|---|---|---|---|
| GRAB | GT | 0.50 | 1.33 | 0 | 0.51 |
| | Input | 4.48 | 5.25 | 13.29 | 881.23 |
| | TOCH | 2.09 | 2.17 | 3.12 | 20.37 |
| | TOCH (w/ MixStyle) | 2.28 | 2.62 | 3.10 | 21.29 |
| | TOCH (w/ Aug.) | 1.94 | 2.04 | 3.16 | 22.58 |
| | Ours (w/o SpatialDiff) | 2.15±0.02 | 2.29±0.03 | 6.71±1.09 | 12.16±0.67 |
| | Ours (w/o TemporalDiff) | **0.86**±0.02 | **1.54**±0.02 | 3.93±0.31 | 9.36±0.68 |
| | Ours (w/o Diffusion) | *1.07* | *1.70* | *2.63* | 10.05 |
| | Ours (w/o Canon.) | 1.57±0.02 | 1.83±0.03 | 2.91±0.28 | *1.30*±0.03 |
| | Ours | 1.22±0.01 | 1.72±0.01 | **2.44**±0.18 | **0.41**±0.01 |
| GRAB (Beta) | GT | 0.50 | 1.33 | 0 | 0.51 |
| | Input | 2.19 | 4.77 | 5.83 | 27.58 |
| | TOCH | 2.33 | 2.77 | 5.60 | 25.05 |
| | TOCH (w/ MixStyle) | 2.01 | 2.63 | 4.65 | 17.37 |
| | TOCH (w/ Aug.) | 1.52 | 1.86 | 3.07 | 13.09 |
| | Ours (w/o Diffusion) | 1.98 | 2.06 | *3.00* | 11.99 |
| | Ours (w/o Canon.) | *1.79*±0.02 | *1.73*±0.03 | 3.19±0.15 | *1.28*±0.03 |
| | Ours | **1.18**±0.00 | **1.69**±0.01 | **2.78**±0.14 | **0.54**±0.00 |
| HOI4D | Input | 2.26 | 2.47 | - | 46.45 |
| | TOCH | 4.09 | 4.46 | - | 35.93 |
| | TOCH (w/ MixStyle) | 4.31 | 4.96 | - | 25.67 |
| | TOCH (w/ Aug.) | 4.20 | 4.51 | - | 25.85 |
| | Ours (w/o Diffusion) | 3.16 | 3.83 | - | 18.65 |
| | Ours (w/o Canon.) | *2.37*±0.02 | *3.57*±0.03 | - | *12.80*±0.79 |
| | Ours | **1.99**±0.02 | **2.14**±0.02 | - | **9.75**±0.88 |
| ARCTIC | GT | 0.33 | 0.92 | 0 | 0.41 |
| | Input | 2.28 | 4.89 | 15.21 | 931.69 |
| | TOCH | 1.84 | 2.01 | 4.31 | 18.50 |
| | TOCH (w/ MixStyle) | 1.92 | 2.13 | *4.25* | *18.02* |
| | TOCH (w/ Aug.) | *1.75* | *1.98* | 5.64 | 22.57 |
| | Ours | **1.35** ± 0.01 | **1.91** ± 0.02 | **2.69** ± 0.11 | **0.85** ± 0.00 |

Table 6: **Quantitative evaluations of the model trained on the ARCTIC training set. Bold red** numbers for best values. "GT" stands for "Ground-Truth".

| Dataset | Method | MPJPE $(mm, \downarrow)$ | MPVPE $(mm, \downarrow)$ | C-IoU $(\%, \uparrow)$ | IV $(cm^3, \downarrow)$ | Penetration Depth $(mm, \downarrow)$ | Proximity Error $(mm, \downarrow)$ | HO Motion Consistency $(mm^2, \downarrow)$ |
|---|---|---|---|---|---|---|---|---|
| GRAB | GT | - | - | - | 0.50 | 1.33 | - | 0.51 |
| | Input | 23.16 | 22.78 | 1.01 | 4.48 | 5.25 | 13.29 | 881.23 |
| | Ours (GRAB) | **9.28** | **9.22** | **25.27** | **1.23** | **1.74** | **2.53** | 0.57 |
| | Ours (ARCTIC) | 11.47 | 11.29 | 24.79 | 1.48 | 1.80 | 2.60 | **0.55** |
| HOI4D | Input | - | - | - | 2.26 | 2.47 | - | 46.45 |
| | Ours (GRAB) | - | - | - | 1.99 | 2.15 | - | 9.81 |
| | Ours (ARCTIC) | - | - | - | **1.54** | **1.96** | - | **9.33** |
| ARCTIC | GT | - | - | - | 0.33 | 0.92 | 0 | 0.41 |
| | Input | 25.51 | 24.84 | 1.68 | 2.28 | 4.89 | 15.21 | 931.69 |
| | Ours (GRAB) | 11.57 | 11.09 | 23.49 | 1.35 | 1.93 | 2.71 | **0.92** |
| | Ours (ARCTIC) | **10.34** | **10.07** | **25.08** | **1.21** | **1.64** | **2.62** | 1.10 |

with correct spatial relations, and to the manifold of trajectories with consistent temporal relations. One may question whether it is possible to use the last projection step only to project the input to the natural interaction manifold in a single step. Our experimental observations show the difficulty of removing such complex noise in one single stage. An effective mapping to clean such complex noise is very hard to learn for neural networks.

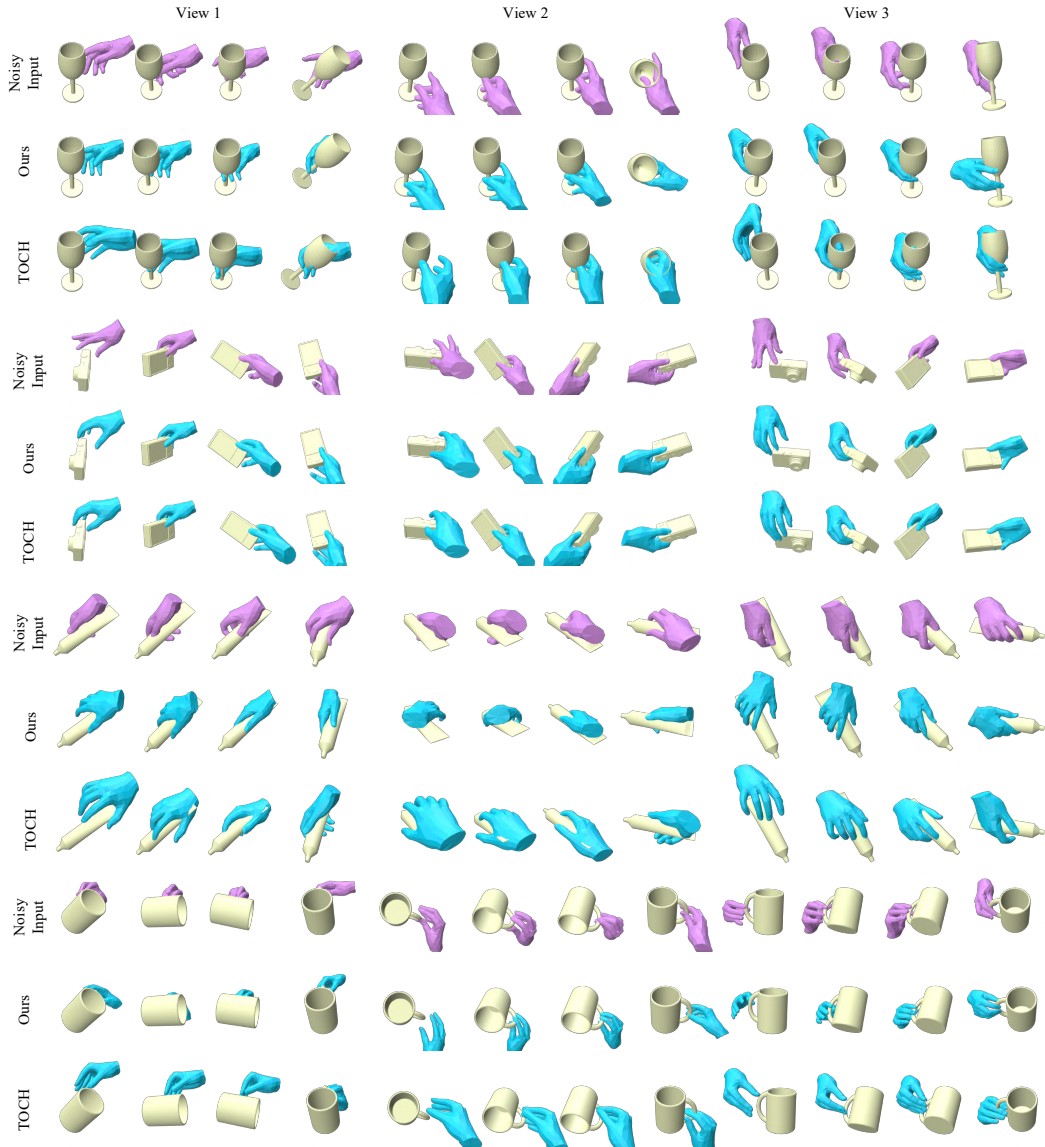

Figure 8: **Evaluation and comparisons on the GRAB test set.** Input and denoised results are shown from three views via four keyframes in the time-increasing order. Please refer to **our website and video** for animated results.

**Generalize from ARCTIC to other datasets.** To further evaluate the generalization ability of our method, we conduct a new series of experiments where we train the model on the ARCTIC dataset (see the Section C for data splitting and other settings) and evaluate on GRAB, HOI4D, and ARCTIC (test split). Table 6 contains its performance. We can observe that though our model trained on the GRAB can generalize to ARCTIC with good performance, the reduced domain gap when using ARCTIC as the training set can really improve the performance. For instance, Figure 15 shows that the model trained on the ARCTIC training set can perform obviously better on examples where the model trained on GRAB would struggle (please see Section B.4 for the discussion on failure case). For the sequence where the hand needs to open wide to hold the microwave, the model trained on GRAB cannot clean the noisy very effectively, producing results with obvious penetrations and the unnatural hand trajectory with instantaneous shaking. However, the model trained on the ARCTIC dataset can eliminate such noise and produce a much natural trajectory. Besides, training on this

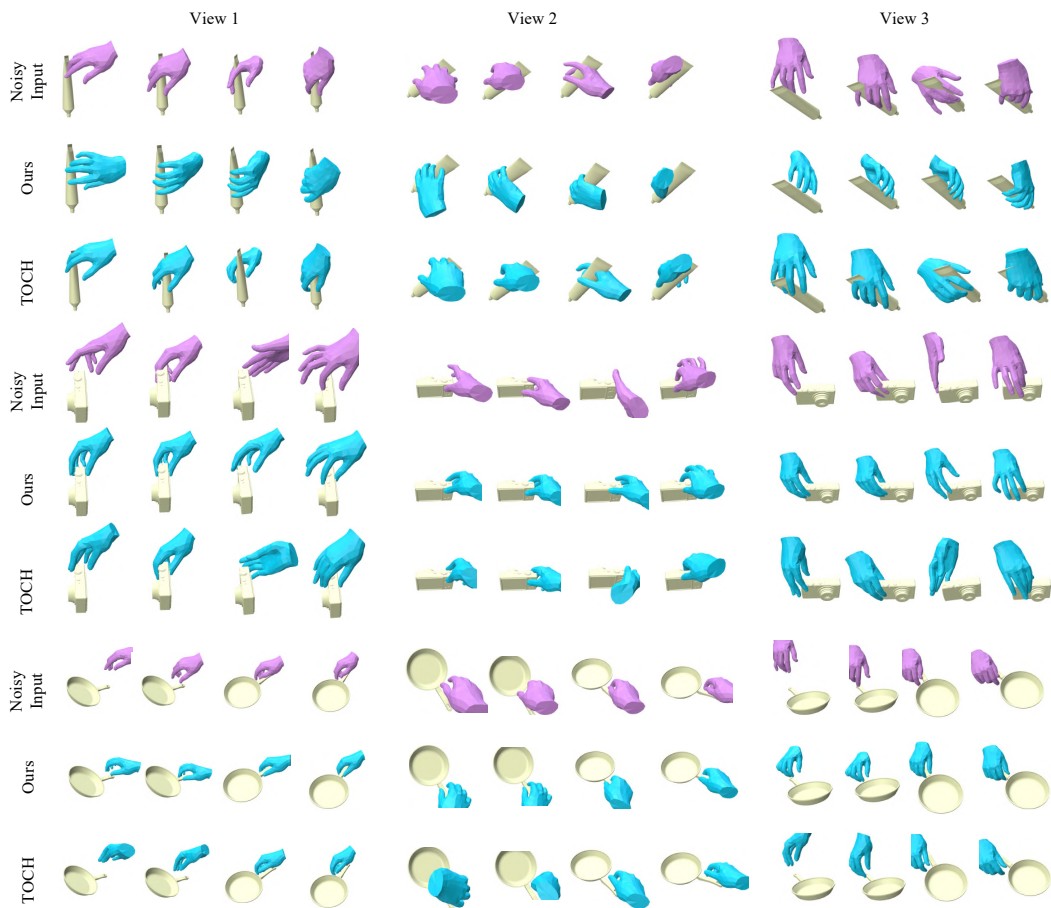

Figure 9: **Evaluation and comparisons on the GRAB (Beta) test set.** Please refer to **our website and video** for animated results.

dataset can benefit the model's performance on the HOI4D dataset with articulated objects and articulated motions.

## B.2 Ablation Studies

This section includes more ablation study results to complement the selected results in the main text. Table 4 and 5 present a more comprehensive quantitative evaluation of ablated models and the comparisons to the full model.

**Generalized contact-centric parameterizations.** Apart from the results present in Table 4 and 5, Figure 16 gives and visual example where the ablated version without such contact-centric design cannot generalize well to the manipulation sequence with large object movements. We can still observe obvious penetrations from all three frames present here.

**Denoising capability on recovering ground-truth and modeling high-frequency pose details.** Together with the ability to model various solutions, the stochastic denoising process also empowers the model to explore a broad space that is more likely to encompass samples close to the ground-truth sequences. Figure 17 shows that ours is more faithful to the ground-truth sequence than the result of TOCH, regarding both recovered hand poses and the contact information.

Besides, taking advantage of the power of our HOI representation GeneOH and the novel denoising scheme, we are able to model high-frequency shape details in the results. However, TOCH's results would exhibit flat hand poses frequently. This may result from its high-dimensional representations

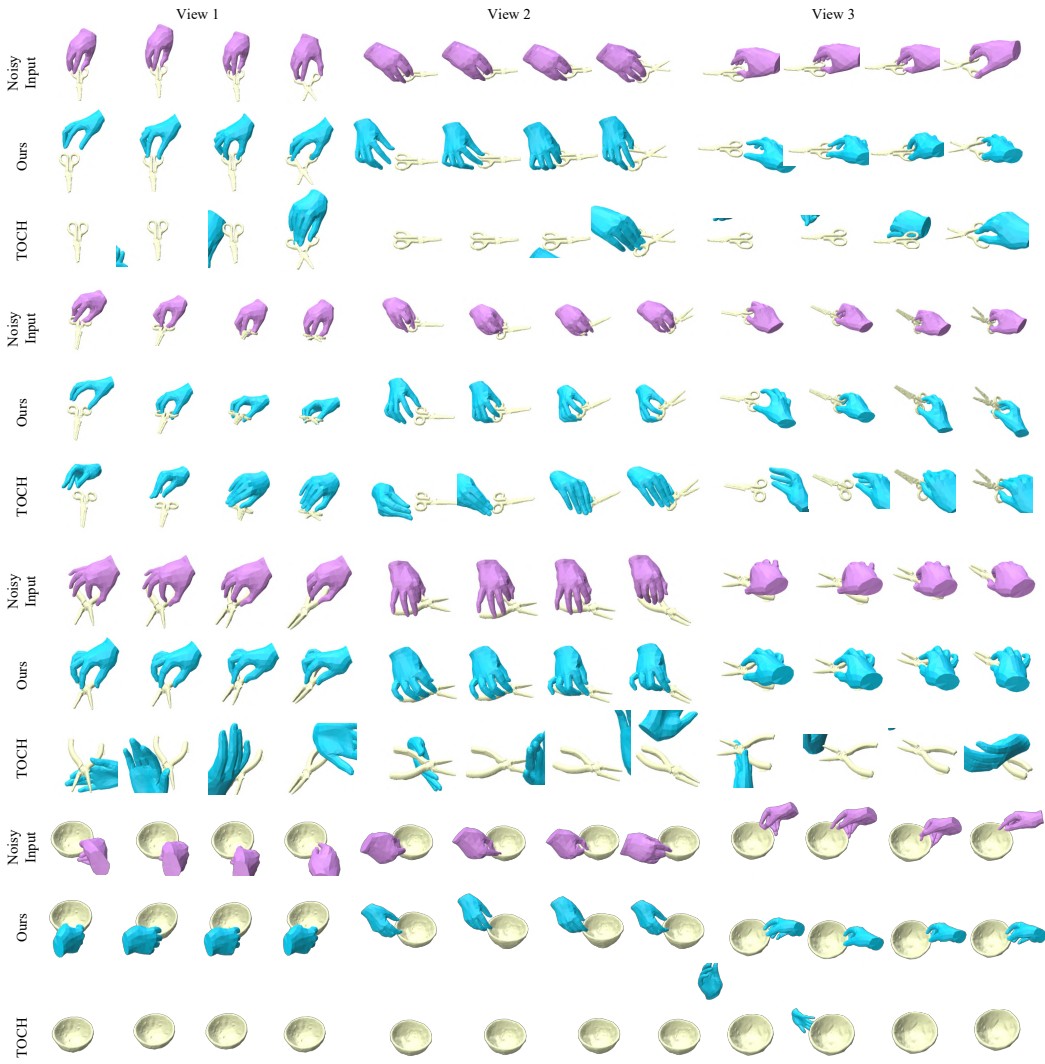

Figure 10: **Evaluation on the HOI4D dataset.** Please refer to **our website and video** for animated results.

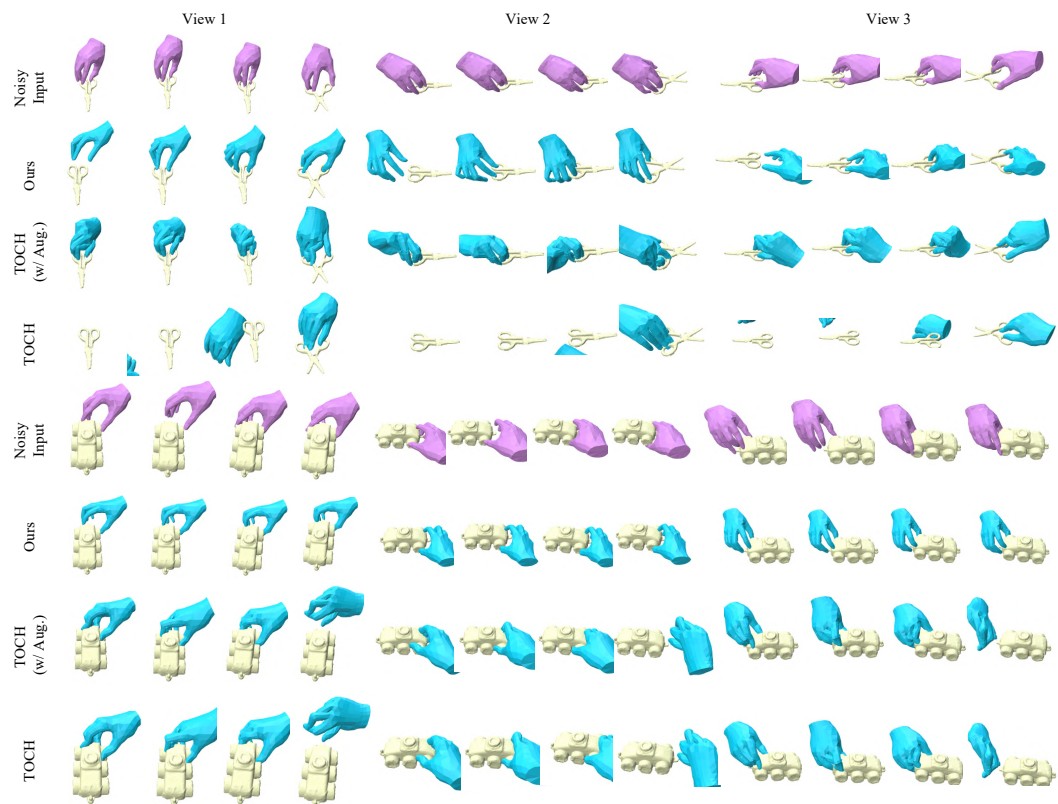

Figure 11: **Comparisons on the HOI4D dataset.** We compare our method with the baseline TOCH and its improved version TOCH (w/ Aug.).

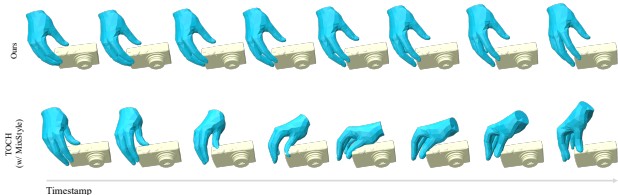

Figure 12: **Weird artifacts produced by TOCH (w/ MixStyle).** (*First line*:) Ours result. (*Second line*:) The result of TOCH (w/ MixStyle). The noisy input is perturbed by noise sampled from a Beta distribution, different from that used in training.

and the limited ability of the denoising strategy, which models the deterministic, one-step noise-to-data mapping relation.

## B.3    APPLICATIONS

This section presents more applications of the denoising model.

**Refining noisy grasps produced by the generation network.** In addition to refining noisy interaction sequences, our method can serve as an effective post-processing tool to refine implausible static grasps produced by the generation network as shown in Figure 6. Examples shown here are grasps taken from interaction results produced by (Wu et al., 2022).

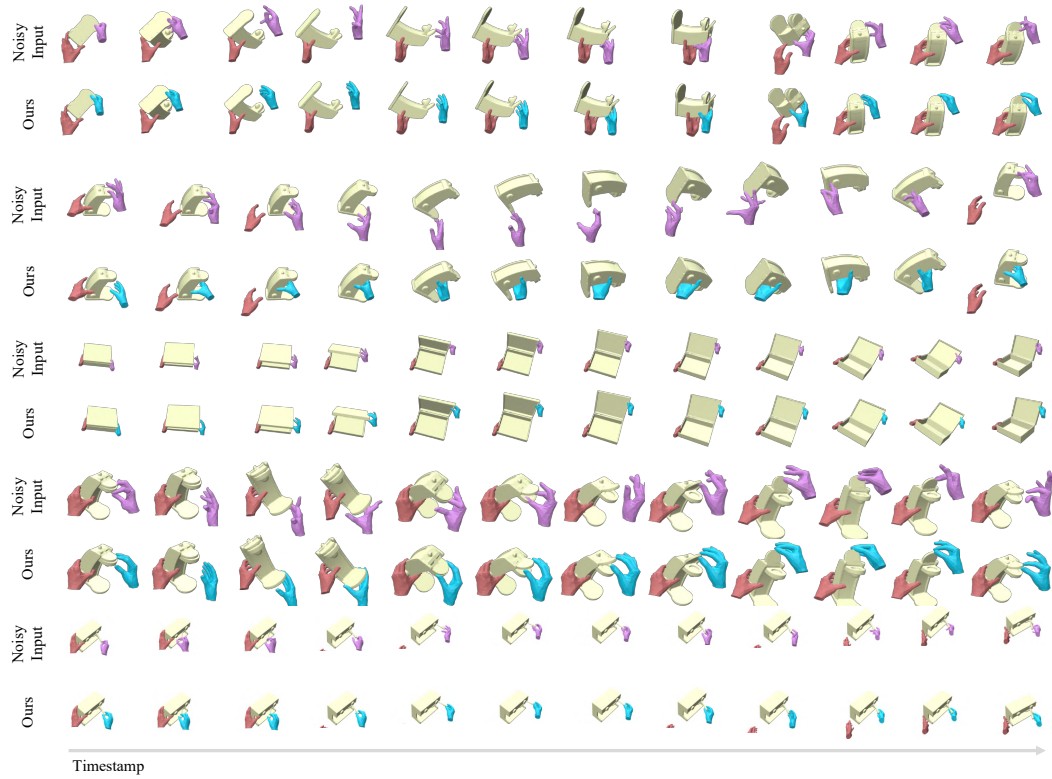

Figure 13: **Evaluation on the ARCTIC dataset.** The model cleans noisy right hand trajectory here. Left hands shown in both the noisy input and the denoised trajectory are GT shapes. Please refer to **our website** and **video** for animated results.

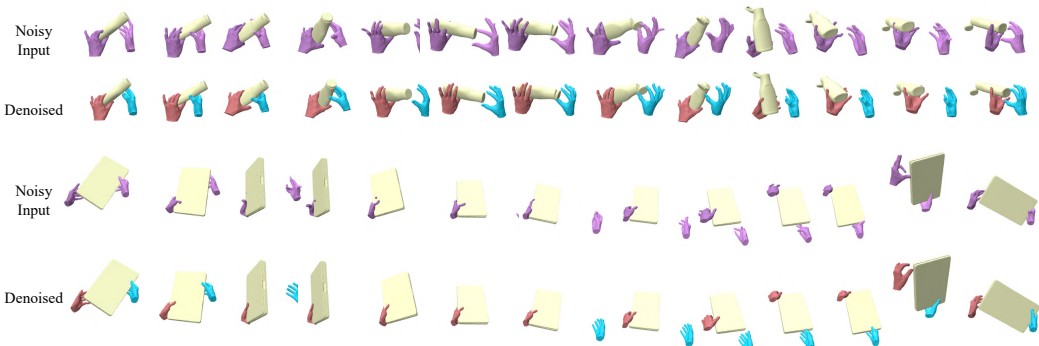

Figure 14: **Evaluation on long interaction sequences with bimanual manipulations.** The model cleans both the noisy right hand trajectory and the noisy left hand trajectory here. Please refer to **our website** and **video** for animated results.

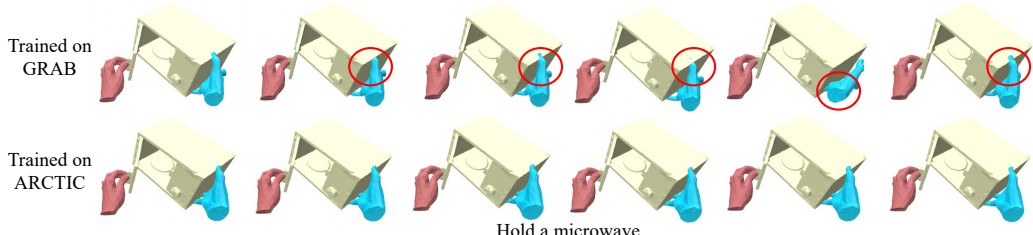

Figure 15: **Comparisons between the model trained on ARCTIC and the one trained on GRAB.** The model trained on ARCTIC training set can generalize to the corresponding test sequences more easily thanks to the reduced domain gap.

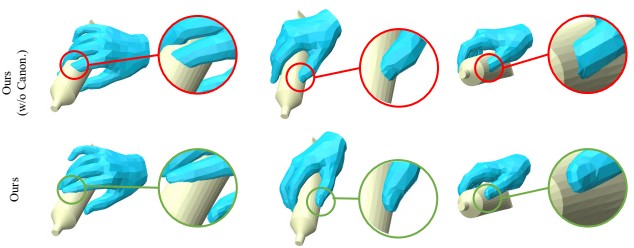

Figure 16: **Effectiveness of the generalized contact-centric parameterization.** (*First line:*) Results of Ours (w/o Canon.). (*Second line:*) Results of our full model.

Apart from the denoising ability, the denoised data with high-quality interaction sequences and static grasps can further aid a variety of downstream tasks. Here we take the grasp synthesis and the manipulation synthesis task as an example.

**Grasp synthesis.** We select four objects and their corresponding grasping poses from the GRAB test set to train the synthesis network. Then we use the network to generate grasps for unseen objects. The results shown in Figure 18 are natural and contact-aware. In contrast, the generated grasps are not plausible as shown in the leftmost part of Figure 18.

**Manipulation synthesis.** We further examine the quality of the denoised interaction data via the manipulation synthesis task. Based on the representations and the network architecture proposed in a recent manipulation synthesis work[1], we train a manipulation synthesis network using our denoised data. The network then takes a new object sequence as input to generate the corresponding manipulation sequence. As shown in Figure 19, the quality of our data is well suited for a learning-based synthesis model. It can generate diverse, high-quality manipulation sequences for an unseen object trajectory.

The above two applications indicate the potential value of our denoising model in aiding high-quality interaction dataset creation.

## B.4 FAILURE CASES

Figure 20 summarizes the failure cases. Our method may sometimes be unable to perform very well in the following situations: 1) When the hand needs to open wide to hold the object, the canonicalized hand trajectories and the hand-object spatial relations canonicalized around the interaction region may be extremely novel to the denoising model. The model then cannot fully clean penetrations from the observations. 2) When the noisy input contains very strange hand motions such as the sudden detachment and grasping presented in Figure 20, the model can remove such artifact but still cannot clean the trajectory perfectly, leaving us remaining penetrations shown in the denoised result. 3) When the hand is opening an unseen object with extremely thin geometry, we may still observe subtle penetrations from the results.

---

[1]https://github.com/cams-hoi/CAMS

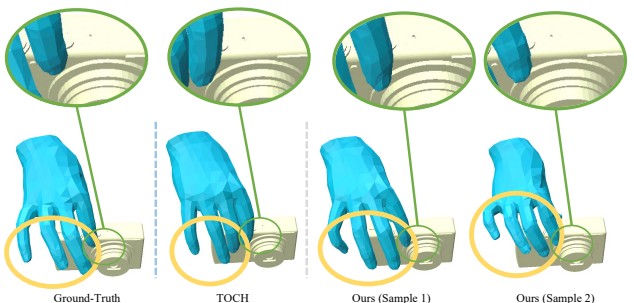

Figure 17: **Comparisons between our model and TOCH on the ability to recover ground-truth interactions.** We can explore a wide space that encompasses the sample with hand poses (highlighted in yellow circles ) and contacts (in green circles) close to the ground truth. We can model high-frequency poses. However, TOCH's result contains plain poses and cannot recover bending fingers exhibited in the ground-truth shape.

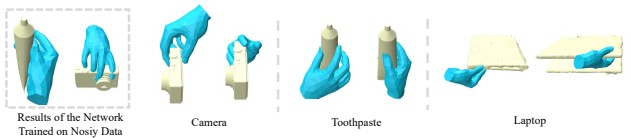

Figure 18: **Grasp synthesis.** Synthesized grasps for unseen objects.

### B.5 ANALYZING THE DISTINCTION BETWEEN NOISE IN REAL HAND-OBJECT INTERACTION TRAJECTORIES AND ARTIFICIAL NOISE

From the visualization and animated results shown on the website and the video, the distinctions between the noise exhibited in real noisy hand-object interaction trajectories (*e.g.,* hand trajectories from the noisy HOI4D dataset and hand trajectories estimated from interaction videos) and the artificial noise could be summarized as follows:

- The trajectories in HOI4D always present unnatural hand poses, jittering motions, missing contacts, and large penetrations;
- Retargeed hand motions always suffer from large penetrations;
- The hand trajectories estimated from HOI videos are usually with penetrations and missing contacts;
- A common feature is that real noisy hand trajectories always present time-consistent artifacts. However, noisy trajectories with artificial noise added independently onto each frame usually present time-varying penetrations and unnatural poses.

Besides, as summarized in Table 1, the differences between different kinds of noise patterns can be revealed by comparing various metrics calculated on their input noisy trajectories, including metrics to reveal penetrations (IV, penetration depth), hand-object proximity (C-IoU, Proximity Error), and motion consistency.

**Further analysis.** We further visualize the difference ($\hat{\theta} - \theta^{gt}$) between noisy hand mano pose parameters ($\hat{\theta}$) and the GT values ($\theta^{gt}$) obtained from the trajectories estimated from videos (the noisy input of the application on cleaning hand trajectory estimations, Sec. 4.4), the difference between the mano pose parameters with artificial Gaussian noise ($\hat{\theta}^{\mathbf{n}}$) and the GT values, and the differences between the parameters with artificial noise drawn from the Beta distribution ($\hat{\theta}^{\mathbf{b}}$). By projecting them into the 2-dimensional plane using the PCA algorithm (implemented in the scikit-learn package), we visualize their positions from 256 examples in Figure 21. As we can see, the real noise pattern is very different from artificial noise. In this case, the noise of the hand trajectory estimated from videos further exhibits instance-specific patterns.

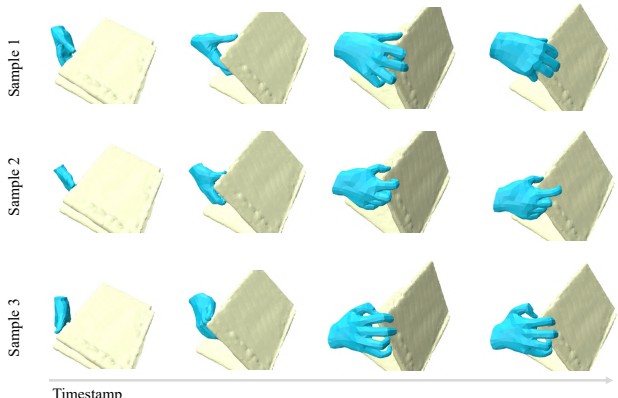

Figure 19: **Manipulation synthesis.** Synthesized manipulation sequences for the unseen laptop object. Frames shown here from left to right are in a time-increasing order.

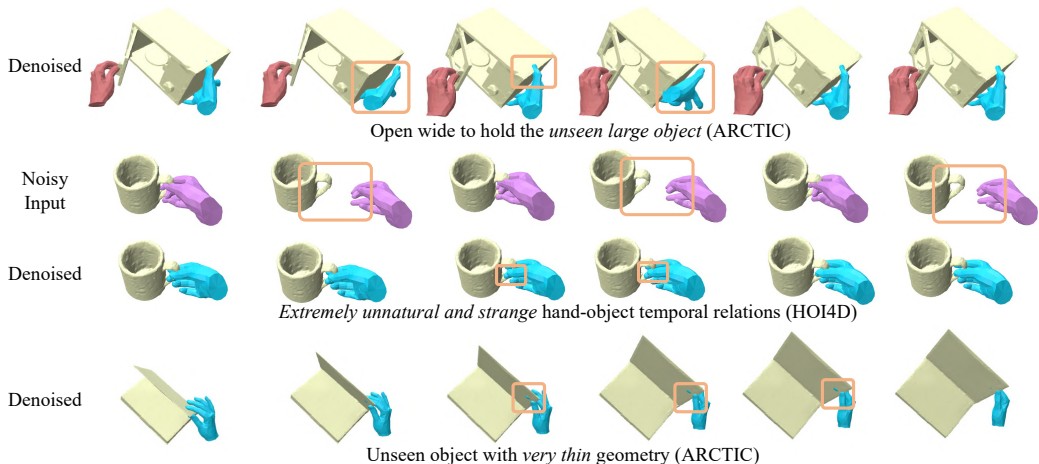

Figure 20: **Failure cases** caused by the *unseen and large object*, *very strange hand-object temporal relations*, and *unseen object with extremely thin geometry*.

### B.6 USER STUDY

To better access and compare the quality of our denoised results to those of the baseline model, we conducted a toy user study. We set up a website containing our denoised results and TOCH's results on 18 noisy trajectories in a randomly permutated order. Twenty people who are not familiar with the task or even have no background in CS are asked to rate each clip a score from 1 to 5, indicating their preferences. Specifically, "1" indicates a significant difference between the hand motion demonstrated in the video and the human behavior, with obvious physical unrealistic phenomena such as penetrations and motion inconsistency; "3" represents the demonstrated motion is plausible and similar to the human behavior, but still suffer from physical artifacts; "5" means a high-quality motion which is plausible with no flaws and is human-like. "2" means the quality is better than "1" but worse than "3". Similarly, "4" means the result is better than "3" but worse than "5".

For each clip, we calculate the average score achieved by our method and TOCH. The average and medium scores across all clips are summarized in Table 7. Ours is much better than the baseline model.

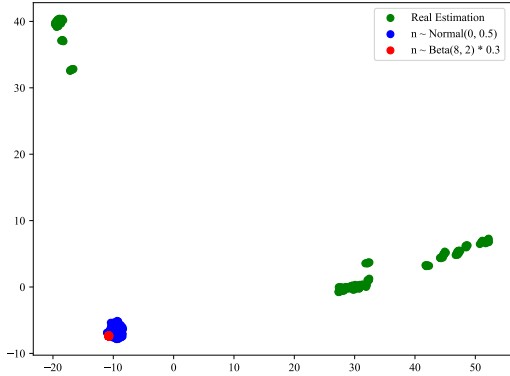

Figure 21: Visualization on the differences between the mano pose parameters of hand trajectories estimated from videos and the GT values, the difference between the mano pose parameters with artificial Gaussian noise ($\hat{\theta}^{\mathbf{n}}$) and the GT values, and the differences between the parameters with artificial noise drawn from the Beta distribution ($\hat{\theta}^{\mathbf{b}}$). The analysis is conducted on 256 trajectories. For each trajectory, the difference vectors across all frames are concatenated together and flattened to a single vector.

Table 7: **User study.**

|  | GeneOH Diffusion | TOCH |
|---|---|---|
| Average Score | **3.96** | 1.98 |
| Medium Score | **4.00** | 1.55 |

## C  EXPERIMENTAL DETAILS

### C.1  DATASETS

**GRAB training set.** This is the training set used in all experiments presented in the **main text**. We follow the cross-object splitting strategy used in (Zhou et al., 2022) to split the GRAB (Taheri et al., 2020) dataset. The training split, containing 1308 manipulation sequences, is used to construct the training dataset. We also filter out frames where the hand wrist is more than 15 cm away from the object. For each training sequence, we slice it into clips with 60 frames to construct the training set. Sequences with a length of less than 60 are not included for training or testing. For models where noisy sequences are required during training, we create the noisy sequence from the clean sequence by adding Gaussian noise to the MANO parameters. Specifically, the Gaussian noise is added to the hand MANO translation, rotation, and pose parameters, with standard deviations of 0.01, 0.1, and 0.5, respectively.

**ARCTIC training dataset.** It is the training set of all models in the experiments where we wish to generalize the model trained on ARCTIC to other datasets. Based on the publicly available sequences from the subject with the index "s01", "s02", "s04", "s05", "s06", "s07", "s08", "s09", "s10", we take the manipulation sequences from "s01" for evaluation and those of other subjects for training. For each sequence, we slice it into small clips with a window size equal to 60 and the step size set to 60. We filter out clips where the maximum distance from the wrist to the nearest object point is larger than 15cm. The number of all training clips is 2524. Only the right hand trajectory is used for training.

The following text contains more details about the four distinct test sets for evaluation, namely GRAB, GRAB (Beta), HOI4D, and ARCTIC.

**GRAB** (Taheri et al., 2020). The test split of the GRAB dataset, containing 246 manipulation sequences, is used to construct the test set. For each test sequence, we slice it into clips with 60 frames using the step size 30. For each test sequence, the noisy sequence is also created by adding Gaus-

sian noise to the MANO parameters. The Gaussian noise is added to the hand MANO translation, rotation, and pose parameters, with standard deviations of 0.01, 0.1, and 0.5, respectively.

**GRAB (Beta).** The GRAB (Beta) test set is constructed from manipulation sequences from the GRAB test split. For each sequence, the noisy sequence is created by adding noise from the Beta distribution ($B(8, 2)$) to the MANO parameters. Specifically, we randomly sample noise from the Beta distribution ($B(8, 2)$). Then the sampled noise was by 0.01, 0.05, 0.3 to get noise vectors added to the translation, rotation, and hand pose parameters respectively.

**HOI4D** (Liu et al., 2022). For the HOI4D dataset, we select interaction sequences with humans manipulating objects from 3 articulated categories, including Laptop, Scissors, and Pliers, and 6 rigid datasets, namely Chair, Bottle, Bowl, Kettle, Mug, and ToyCar, for test. The number of instances included in each category is detailed in Table 3. For each sequence, we specify the starting frame and take the clip with the length of 60 frames starting from it as the test clip. The starting frame set for each category is listed in Table 3.

**ARCTIC** (Fan et al., 2023). The ARCTIC test set for evaluation takes the right hand only since we observe that dexterous manipulation such as articulated manipulations is always conducted by the right hand. For instance, as shown in the example in Figure 4, the left hand holds the capsule machine with no contact change during the manipulation while the right hand first touches the lid, then touches the base, and then opens and close the lid. *However, we can refine the left hand motions as well*, as demonstrated in the second sequence of the "refining estimation from video" example shown in Figure 1. The manipulation sequences from "s01", 34 in total, are taken for evaluation. For each test sequence, the quantitative results are evaluated from clips with the window size 60 sliced from each sequence using the step size 30. The filtering strategy similar to that used for constructing the training dataset is applied to the test clips as well. The default length of the clips used in the qualitative evaluation is 90, which is the composed result of two adjacent clips with a window size of 60. The noisy sequence is obtained by adding Gaussian noise to the right hand MANO parameters. Specifically, the Gaussian noise is added to the hand MANO translation, rotation, and pose parameters, with standard deviations of 0.01, 0.05, and 0.3, respectively.

Since the ARCTIC's object template meshes do not provide vertex normals, which are demanded both in our method and some baseline models, we use the "compute_vertex_normals" function implemented in Open3D (Zhou et al., 2018) for computing the vertex normals.

## C.2 METRICS

We include two sets of evaluation metrics. The first set follows the evaluation protocol of previous works (Zhou et al., 2022) and focuses on assessing the model's capability to recover the GT trajectories from noisy observations, as detailed in the following.

**Mean Per-Joint Position Error (MPJPE).** It calculates the average Euclidean distance between the denoised 3D hand joints and the corresponding ground-truth joints.

**Mean Per-Vertex Position Error (MPVPE).** It measures the average Euclidean distance between the denoised 3D hand vertices and the corresponding ground-truth vertices.

**Contact IoU (C-IoU).** This metric assesses the similarity between the refined contact map and the ground-truth contact map. The binary contact maps are obtained by thresholding the correspondence distance within ±2mm. For our method, which does not rely on correspondences introduced in (Zhou et al., 2022), we utilize the computing process provided by (Zhou et al., 2022) to compute the correspondences.

To measure whether the denoised trajectory exhibits natural hand-object spatial relations and consistent hand-object motions, we introduce the second set of evaluation metrics.

**Solid Intersection Volume (IV).** We evaluate this metric following (Zhou et al., 2022). It quantifies hand-object inter-penetrations. By voxelizing the hand mesh and the object mesh, we calculate the volume of their intersected region as the intersection volume.

**Per-Vertex Maximum Penetration Depth (Penetration Depth).** For each frame, we calculate the maximum penetration depth of each hand vertex into the object. We then average these values across all frames to obtain the per-vertex maximum penetration depth.

**Proximity Error.** The metric is only evaluated on datasets with ground-truth references, including GRAB, GRAB (Beta), and ARCTIC. For each vertex of the denoised hand mesh, we compute the difference between its minimum distance to the object points and the corresponding ground-truth vertex's minimum distance to the object points. The proximity error is obtained by averaging these differences over all vertices. The overall metric is obtained by averaging the per-frame metric over all frames. Specifically, let $d_{k,min}^{\mathbf{h}}$ denote the minimum distance from the hand keypoint $\mathbf{h}_k$ at the frame $k$ to objects points. Formally, it is defined as $d_{k,min}^{\mathbf{h}} = \min\{d_k^{\mathbf{ho}} = \|\mathbf{h}_k - \mathbf{o}_k\|_2 | \mathbf{h}_k \in \mathbf{J}_k, \mathbf{o}_k \in \mathbf{P}_k\}$. Let $d_{k,min}^{\hat{\mathbf{h}}}$ represents the quantity of the keypoint $\mathbf{h}$ from the denoised trajectory, and $d_{k,min}^{\mathbf{h}}$ represents the quantity of the keypoint $\mathbf{h}$ from the ground-truth trajectory. Then the overall metric is calculated as Proximity error $= \text{mean}\{\{\|d_{k,min}^{\hat{\mathbf{h}}} - d_{k,min}^{\mathbf{h}}\|_2 | \mathbf{h}_k \in \mathbf{J}_k\} | 1 \leq k \leq K\}$.

**Hand-Object Motion Consistency (HO Motion Consistency).** This metric assesses the consistency between the hand and object motions. For each frame where the object is not static, we identify the nearest hand-object point pair $(\mathbf{h}_k, \mathbf{o}_k) = \text{argmin}\{d_k^{\mathbf{ho}} | (\mathbf{h}_k \in \mathbf{J}_k, \mathbf{o}_k \in \mathbf{P}_k)\}$. We use the expression $\|e^{-100\|\mathbf{h}_k - \mathbf{o}_k\|_2}\Delta\mathbf{h}_k - \Delta\mathbf{o}_k\|_2^2$ to quantify the level of inconsistency between the hand and object motions. Here, $\Delta\mathbf{h}_k$ and $\Delta\mathbf{o}_k$ represent the displacements of the hand point and the object point between adjacent frames, respectively. We obtain the overall metric by averaging the metric over all frames.

## C.3 BASELINES

We give a more detailed explanation of the compared baselines as follows to complement the brief introduction in the main text.

**TOCH.** We compare our model with the prior art on the HOI denoising problem, TOCH (Zhou et al., 2022). TOCH utilizes an autoencoder structure and learns to map noisy trajectories to their corresponding clean trajectories. By projecting input noisy trajectories onto the clean data manifold, it can accomplish the denoising task. We utilize the official code provided by the authors for training and evaluation.

**TOCH (w/ MixStyle).** Further, to improve TOCH's generalization ability towards new interactions, we augment it with a general domain generalization method MixStyle (Zhou et al., 2021a), resulting in a variant named "TOCH (w/ MixStyle)".

**TOCH (w/ Aug.).** Another variant, "TOCH (w/ Aug.)", where TOCH is trained on the training sets of the GRAB and GRAB (Beta) datasets, is further introduced to enhance its robustness towards unseen noise patterns.

## C.4 MODELS

**Denoising models used in our method.** We realize the denoising model's function and the training as those of the score functions in diffusion-based generative models. We adapt the implementation of Human Motion Diffusion (Tevet et al., 2022) to implement our three denoising models for each part of the representation[2]. Instead of training the denoising function to predict the start data point $\mathbf{x}_0$ from the noisy data $\mathbf{x}_t$ as implemented in (Tevet et al., 2022), we predict the noise $(\mathbf{x}_t - \mathbf{x}_0)$. We also follow its default training protocol.

We mainly adopt MLPs and Transformers as the basic backbones of the denoising model. The detailed structure depends on the type of the corresponding statistics and the dimensions. **The code in the Supplementary Materials provides all those details.** So we spare the effort to list them in detail here.

When leveraging the denoising model to clean the input via the "denoising via diffusion" strategy, the diffusion steps is set to 400 for MotionDiff, 200 for SpatialDiff, and 100 for TemporalDiff empirically.

**Ours (w/o Diffusion).** In this ablated version, we design a denoising autoencoder for cleaning the spatial and temporal representations. For each representation $\bar{\mathcal{J}}, \mathcal{S}, \mathcal{T}$, we leverage an autoencoder for denoising. After that, we get the final hand meshes by fitting the MANO parameters

---

[2]https://guytevet.github.io/mdm-page

$\{\mathbf{r}_k, \mathbf{t}_k, \beta_k, \theta_k\}$ to reconstruct the denoised representations. Assuming the reconstructed hand trajectory as $\mathcal{J}^{recon}$, the reconstructed hand-object spatial relative positions as $\mathcal{S}^{recon}$, and the temporal representations as $\mathcal{T}^{recon}$, the reconstruction loss is formulated as follows:

$$\mathcal{L}_{recon}^{rep} = \lambda_1 \|\mathcal{J} - \mathcal{J}^{recon}\|_2 + \lambda_2 \|\mathcal{S} - \mathcal{S}^{recon}\| + \lambda_3 \|\mathcal{T} - \mathcal{T}^{recon}\|, \tag{14}$$

where $\lambda_1, \lambda_2, \lambda_3$ are coefficients for the reconstruction losses. We set $\lambda_1, \lambda_2, \lambda_3 = 1.$ in our experiments. The distance function between the spatial representations is calculated on the relative positions between each point pair. The distance between the temporal representations is calculated on hand-object distances, *i.e.*, $\{d_k^{\mathbf{ho}}\}$, and two relative velocity-related statistics ($\{e_{k,\|}^{\mathbf{ho}}, e_{k,\perp}^{\mathbf{ho}}\}$). Together with the regularization loss

$$\mathcal{L}_{reg} = \frac{1}{K} \sum_{k=1}^{K} (\|\beta_k\|_2 + \|\theta_k\|_2) + \frac{1}{K-1} \sum_{k=1}^{K-1} \|\theta_{k+1} - \theta_k\|_2, \tag{15}$$

the total optimization target is formulated as follows,

$$\text{minimize}_{\{\mathbf{r}_k, \mathbf{t}_k, \beta_k, \theta_k\}_{k=1}^{K}} (\mathcal{L}_{recon}^{rep} + \mathcal{L}_{reg}), \tag{16}$$

and we employ an Adam optimizer to solve the problem.

**TOCH (w/ MixStyle).** We use the official code provided to implement the MixStyle layer. We add a MixStyle layer between every two encoder layers of the TOCH model (Zhou et al., 2022). Configurations of MixStyle are kept the same as the default setting.

**TOCH (w/ Aug.).** The model is trained on paired noisy-clean data pair from the GRAB training set. We perturb each training sequence with two types of noise, that is the noise from a Gaussian distribution, and the noise from a Beta $B(8,2)$ distribution. The noise scale for the Gaussian for the translation, rotation, and hand poses are 0.01, 0.1, and 0.5 respectively. The scale of the Beta noise added on the translation, rotation, and hand poses are 0.01, 0.05, and 0.3.

**Grasp synthesis network.** We adapt the WholeGrasp-VAE network proposed in (Wu et al., 2022) to a HandGrasp-VAE network[3]. Instead of using whole-body markers, we use hand anchor points (Yang et al., 2021), composed of 32 points from the hand palm in total. To identify contact maps for both hand anchors and object points, we set a distance threshold, *i.e.,* 2 mm, and mark the status of points with the minimum distance to the hand/object as contact. During training, we do not add the ground contact loss since the whole body is not considered in our hand-grasping setting. To train the network, we further split the GRAB test set into a subset containing binoculars, wineglass, fryingpan, and mug for training. Then we use the network to synthesize grasps for unseen objects. We select 100 grasps for each object to construct the training dataset.

**Manipulation synthesis network.** We utilize the denoised manipulation trajectories for the Laptop category to train the manipulation synthesis network. The training data consists of 100 manipulation sequences.

## C.5 TRAINING AND EVALUATION

**The denoising model for $\bar{\mathcal{J}}$.** The denoising model for the canonicalized hand trajectory $\bar{\mathcal{J}}$ is trained on canonicalized hand trajectories $\{\bar{\mathcal{J}}\}$ of all interaction sequences in the training set. We apply per-instance normalization operation to those points at each frame for centralization and scaling purposes. Specifically, we utilize the mean and the standard deviation statistics calculated for all points across all frames. In more detail, we first concatenate all keypoints over all frames to form the concatenated keypoints $\mathbf{J}^{concat}$:

$$\bar{\mathbf{J}}_{\text{concat}} = \text{Concat}\{\bar{\mathbf{J}}_k, \dim = 0\}_{k=1}^{K}, \tag{17}$$

where $\bar{\mathbf{J}}_k \in \mathbb{R}^{N_h \times 3}$ for each frame $k$. Then, the average and the standard deviation is calculated on $\bar{\mathbf{J}}^{\text{concat}}$ via

$$\mu^{\bar{\mathbf{J}}} = \text{Average}(\bar{\mathbf{J}}_{\text{concat}}, \dim = 0) \tag{18}$$

$$\sigma^{\bar{\mathbf{J}}} = \text{Std}(\bar{\mathbf{J}}_{\text{concat}}, \dim = 0). \tag{19}$$

---

[3]https://github.com/JiahaoPlus/SAGA

The $(\mu^{\bar{\mathbf{J}}}, \sigma^{\bar{\mathbf{J}}})$ are utilized to normalize $\bar{\mathbf{J}}_k$ at each frame $k$, *i.e.,*

$$\bar{\mathbf{J}}_k \leftarrow \frac{\bar{\mathbf{J}}_k - \mu^{\bar{\mathbf{J}}}}{\sigma^{\bar{\mathbf{J}}}}. \tag{20}$$

**The denoising model for $\mathcal{S}$.** Similarly, the denoising model for hand-object spatial relations $\mathcal{S}$ is trained using representations $\{\mathcal{S}\}$ from all interaction sequences in the training set. We apply per-instance normalization to the canonicalized hand-object relative positions $\{(\mathbf{h}_k - \mathbf{o}_k)\mathbf{R}_k^T\}$. The normalization is conducted in a per-instance per-object point way. For each object point $\mathbf{o}$ in the generalized contact points $\mathbf{P}$, we calculate the average and standard deviation of $\{\{\mathbf{h}_k - \mathbf{o}_k\}\}$ over all frames $k$. We first concatenate the relative positions over all frames and all hand keypoints for the concatenated spatial relations, denoted as

$$\mathbf{s}^{\mathbf{o}}_{\text{concat}} = \text{Concat}\{\{\mathbf{h}_k - \mathbf{o}_k\}, \dim = 0\}_{k=1}^K. \tag{21}$$

Then, the average and the standard deviation is calculated on $\mathbf{s}^{\mathbf{o}}_{\text{concat}}$ via

$$\mu^{\mathbf{o}} = \text{Average}(\mathbf{s}^{\mathbf{o}}_{\text{concat}}, \dim = 0) \tag{22}$$
$$\sigma^{\mathbf{o}} = \text{Std}(\mathbf{s}^{\mathbf{o}}_{\text{concat}}, \dim = 0). \tag{23}$$

Such statistics $(\mu^{\mathbf{o}}, \sigma^{\mathbf{o}})$ are utilized to normalize the relative positions $\{\{\mathbf{h}_k - \mathbf{o}_k\}\}$, *i.e.,*

$$(\mathbf{h}_k - \mathbf{o}_k) \leftarrow \frac{(\mathbf{h}_k - \mathbf{o}_k) - \mu^{\mathbf{o}}}{\sigma^{\mathbf{o}}}. \tag{24}$$

**The denoising model for $\mathcal{T}$.** When training the denoising model for the hand-object temporal relations $\mathcal{T}$, we first train an autoencoder, composed of an $\text{encode}(\cdot)$ function and a $\text{decode}(\cdot)$ function for $\mathcal{T}$. It takes the $\mathcal{T}$ as input and decode the hand-object distances $\{d_k^{\mathbf{ho}}\}$ and the relative velocity-related quantities $\{e_{k,\perp}^{\mathbf{ho}}, e_{k,\parallel}^{\mathbf{ho}}\}$. Then, the denoising model is trained on the encoded latent $\{\text{encode}(\mathcal{T})\}$. This approach avoids the need for designing normalization strategies for the temporal representations. We adopt a PointNet structure block with a positional encoder followed by a transformer encoder module for encoding the temporal relation representations. Given the input temporal representation $\hat{T} \in \mathbb{R}^{K \times N_o \times 69}$, the PointNet encoder block passes it through four PointNet blocks each with three encoding layers with latent dimension $(32, 32, 32), (64, 64, 64), (128, 128, 128), (256, 256, 256)$ respectively. The transformer encoder module is with parameters "num_heads" as 4, feedforward latent dimension as 1024, dropout rate 0, and the latent dimension 256. The decoder contains fully connected layers for decoding each kind of statistics $d_k^{\mathbf{ho}}, e_{\perp,k}^{\mathbf{ho}}, e_{\parallel,k}^{\mathbf{ho}}$ individually.

**Train-time rotation augmentation.** For the canonicalized hand trajectory representation $\bar{\mathcal{J}}$, the train-time random rotation augmentation applies a single random rotation matrix to the whole canonicalized hand trajectories. The same random rotation matrix, denoted as $\mathbf{R}_{\text{rnd}}$, is added to the hand-object spatial representation $\mathcal{S}$ as well. It is used to transform the canonicalized object position, normal, and the hand-object offset vector:

$$\mathbf{s}^{\mathbf{o}}_k \mathbf{R}_{\text{rnd}} = ((\mathbf{o}_k - \mathbf{t}_k)\mathbf{R}_k^T \mathbf{R}_{\text{rnd}}, \mathbf{n}_k \mathbf{R}_k^T \mathbf{R}_{\text{rnd}}, \{(\mathbf{h}_k - \mathbf{o}_k)\mathbf{R}_k^T \mathbf{R}_{\text{rnd}} | \mathbf{h}_k \in \mathbf{J}_k\}). \tag{25}$$

Similarly, the same random rotation matrix $\mathbf{R}_{\text{rnd}}$ is used to transform the object velocity vector from $\mathbf{v}_k^{\mathbf{o}}$ in the temporal representation $\mathcal{T}$ to $\mathbf{v}_k^{\mathbf{o}} \mathbf{R}_{\text{rnd}}$.

### C.6   COMPLEXITY AND RUNNING TIME DISCUSSION

Denote the number of hand keypoints as $|\mathcal{J}|$, the number of generalized contact points $|\mathcal{P}|$, the complexity, the average inference time, and the number of forward diffusion steps for each denoising stage during inference are summarized in Table 8.

Table 8: **Complexity and running time during the inference time.**

|  | MotionDiff | SpatialDiff | TemporalDiff |
|---|---|---|---|
| Average inference time (s) | 0.52 | 16.61 | 7.04 |
| Complexity | $\mathcal{O}(\lvert\mathcal{J}\rvert)$ | $\mathcal{O}(\lvert\mathcal{P}\rvert\lvert\mathcal{J}\rvert)$ | $\mathcal{O}(\lvert\mathcal{P}\rvert\lvert\mathcal{J}\rvert)$ |
| #Forword diffusion steps | 400 | 200 | 100 |

