# OpenReview forum: "GeneOH Diffusion: Towards Generalizable Hand-Object Interaction Denoising via Denoising Diffusion"
_ICLR.cc/2024/Conference — ICLR 2024 poster_

### Official Review · Reviewer_A9wU · 2023-10-29

**Soundness:** 3 good
**Presentation:** 3 good
**Contribution:** 3 good
**Rating:** 6
**Confidence:** 4

**Summary:**

This paper tackles the problem of denoising HOI in images. The proposed method, GeneOH Diffusion, uses a novel approach that includes a contact-centric HOI representation named GeneOH and a new domain-generalizable denoising scheme. The contact-centric representation GeneOH parameterizes the HOI process, facilitating enhanced generalization across various HOI scenarios. The new denoising scheme consists of a canonical denoising model trained to project noisy data samples from a whitened noise space to a clean data manifold and a denoising via diffusion. The experiments on four benchmarks show the superior effectiveness of the proposed method.

**Strengths:**

* GeneOH that encodes hand trajectories, hand-object spatial relations, and hand-object temporal relations to faithfully capture the interaction process.
* Extensive experiments on four benchmarks with significant domain variations show the effectiveness of the method.
* Codes are provided in the supplementary materials which is good for reproduction.

**Weaknesses:**

* The framework is complicated with three denoising stages. Complexity and running time discussion is missing in this paper.

* Comparison and discussion with more recent works are missing, e.g. reference [1] is also a diffusion based HOI method.

* Whitened noise space is constructed by diffusing the training data towards a random Gaussian noise. However, this technique is a common in diffusion model based applications.

* No user studies or applications presented to demonstrate the practical utility of the denoised results. Quantitative metrics alone do not prove the outputs are naturally refined.


[1] Diffusion-Guided Reconstruction of Everyday Hand-Object Interaction Clips

**Questions:**

See the weaknesses part.

---

> ### Author Response · Authors · 2023-11-16
> **Author Response (Part 1 of 3)**
>
> Dear Reviewer A9wU,
>
> Thank you sincerely for your careful and constructive review. We appreciate your recognition of the GeneOH representation design, the comprehensive evaluations, and the demonstrated effectiveness of our work. In the following text, we provide responses to your questions. Hope that they can address your concerns.
>
>
>
> ### The technical novelties and contributions of our diffusion-based denoising technique
>
> First, please allow us to briefly summarize and emphasize the core technical contributions and novelties of our method for HOI denoising.
>
> **Aligning to the Whitened Noise Space for Generalizable Denoising:**
>
> - The whitened noise space, though constructed by diffusing training data towards a random Gaussian noise, is not used for drawing new samples from the same distribution as in standard approaches.
> - Instead, it serves as an intermediate to bridge noisy input from distinct distributions for generalizable denoising, addressing the over-fitting issue present in previous methods trained on noisy data with specific types of noise.
> - Our novel denoising paradigm successfully provides a significant advancement over traditional schemes, which typically fail to tackle the challenging generalizable denoising problem.
>
> **Progressively Cleaning Difficult and Heterogeneous HOI Noise:**
>
> - We introduce a progressive distribution modeling and manifold mapping strategy for training the denoising model, addressing the complex HOI denoising problem.
> - Our stage-wise denoising approach successfully cleans complex noise progressively. Each stage operates harmoniously without undermining the denoised results achieved by previous stages.
>
> **Novel GeneOH Representation:**
>
> - Our method introduces the GeneOH representation, encoding the HOI process informatively, and surpassing prior designs, as illustrated in the last paragraph of Sec. 3.1 and Appendix A.1.
>
> **In Summary:** The core contributions of our method, including the innovative use of diffusion-based techniques for modeling complex HOI distributions and achieving generalizable denoising, along with the design of the GeneOH representation, represent highly non-trivial advancements over traditional paradigms.
>
>
>
> ### "The framework is complicated...Complexity and running time discussion is missing..."
>
> - The intricate framework featuring three denoising stages is imperative for addressing the intricate challenge of HOI denoising. A method lacking careful and intricate design often falls short in effectively cleaning complex and heterogeneous HOI noise.
> - Each denoising stage in our method plays a crucial role and solidly contributes to our overall performance, as demonstrated in our ablation studies (both in paper and in the animated results on the [website](https://geneoh-diffusion.github.io/Geneoh-Diffusion/) and the [supplementary video](https://geneoh-diffusion.github.io/Geneoh-Diffusion/static/videos/video-2.mp4)).
>
> **Complexity and Running Time Discussion:**
>
> Denoting the number of hand keypoints as $\vert\mathcal{J}\vert$ and the number of generalized contact points as $\vert \mathcal{P}\vert$, the complexity, average running time, and the number of forward diffusion steps for each denoising stage ***during inference*** are summarized as follows:
>
> |                            | MotionDiff                             | SpatialDiff                                                  | TemporalDiff                                                 |
> | -------------------------- | -------------------------------------- | ------------------------------------------------------------ | ------------------------------------------------------------ |
> | Average inference time (s) | 0.52                                   | 16.61                                                        | 7.04                                                         |
> | Complexity                 | $\mathcal{O}(\vert \mathcal{J} \vert)$ | $\mathcal{O}(\vert \mathcal{P} \vert \vert \mathcal{J} \vert)$ | $\mathcal{O}(\vert \mathcal{P} \vert \vert \mathcal{J} \vert)$ |
> | #Forword diffusion steps   | 400                                    | 200                                                          | 100                                                          |
>
> We have integrated them in the *Appendix C.6* in the revision.

---

> ### Author Response · Authors · 2023-11-16
> **Author Response (Part 2 of 3)**
>
> ### Differences between our method and the DiffHOI [1]
>
> - Initially, we did not include a discussion or comparison with DiffHOI since its focus and contribution are almost orthogonal to ours. DiffHOI aims to reconstruct better object mesh sequences from molecular videos and, therefore, ***cannot solve our problem***. It does not emphasize the quality of interaction reconstruction, making direct comparisons inappropriate:
>   - As stated in the first sentence of its Sec. 4.2, **Evaluation Metric**, "We evaluate the ***object reconstruction errors***," highlighting DiffHOI's focus on object reconstruction quality.
>   - Visualizations in the paper and on its [website](https://judyye.github.io/diffhoi-www/) reveal noticeable motion jittering in their hand trajectories and implausible hand-object relations, such as penetrations and missing contacts.
>   - HOI4D dataset serves as a data source that offers us noisy trajectories with real noise patterns. We show our denoising capability on this dataset by showing that the trajectories denoised by our method present a significant improvement regarding the physical validity over the original data. However, DiffHOI treats the original annotations as accurate and evaluates their method by comparing their reconstructions to them.
>
> - In our work, we focus on achieving high-quality and motion-aware HOI denoising. The resulting hand trajectory should exhibit natural hand poses and interact with the object in a spatially and temporally plausible manner. None of these aspects align with DiffHOI's objectives, and our results, available [here](https://geneoh-diffusion.github.io/Geneoh-Diffusion/), demonstrate significantly higher quality regarding plausible interactions.
>
> - Rather than treating DiffHOI as closely related work, it seems more promising to leverage our work as a post-processing tool to refine DiffHOI's reconstructions. However, this experiment cannot be conducted at present as ***their code or raw reconstructions are not publicly released***.
>
> - We have added a discussion on this reference in the revision (*Sec. 2*).

---

> ### Author Response · Authors · 2023-11-16
> **Author Response (Part 3 of 3)**
>
> ### "No user studies or applications...practical utility...results. Quantitative...do not prove...are naturally refined."
>
> **Applications Demonstrating Practical Utility:**
>
> We wish to clarify that our paper encompasses a diverse range of applications, thoroughly discussed in *Sec. 4.2, Sec. 4.4,* and *Appendix B.3*. These applications include: 1) Cleaning real noisy datasets (HOI4D); 2) Refining motion-retargeted trajectories; 3) Enhancing trajectories estimated from HOI videos; 4) Refining noisy grasps generated by the synthesis network; 5) Serving as an effective tool for dataset creation in synthesis networks.
>
> Our method's performance across these applications serves as a testament to its high generalization ability and practical utility. Taking the denoising of HOI4D as an example, our approach excels in challenging cases where previous methods struggle. This success is evident in Table 1, Figure 4, 10, and 11, as well as in the animated results available on our [website](https://geneoh-diffusion.github.io/Geneoh-Diffusion/) and [video](https://geneoh-diffusion.github.io/Geneoh-Diffusion/static/videos/video-2.mp4).
>
> **Qualitative Evaluations:**
> In addition to quantitative assessments, our paper features numerous qualitative evaluations to provide a comprehensive understanding of the quality of our denoised results. These include visualizations in Figure 4, 5, 8, 9, 10, 11, 12, 13, and 14 within the paper, as well as animated results available on our [website](https://geneoh-diffusion.github.io/Geneoh-Diffusion/) and in the [supplementary video](https://geneoh-diffusion.github.io/Geneoh-Diffusion/static/videos/video-2.mp4). These qualitative analyses aim to offer a nuanced perspective on the effectiveness and realism of our denoising outcomes.
>
> **User Study:**
>
> To better assess and compare the quality of our denoised results with those of the baseline model, we conducted a toy user study:
>
> - We set up a website presenting our denoised results alongside TOCH's results for 18 noisy trajectories.
> - Twenty individuals, with no prior familiarity with the task and varying backgrounds, were asked to rate each clip a score from 1 to 5. The score reflects their preferences: `1` indicates significant differences between the demonstrated hand motion and human behavior, with noticeable physical unrealistic phenomena like penetrations and motion inconsistency. `3` signifies plausible motion with some artifacts, and `5` represents high-quality, flaw-free, and human-like motion. Scores `2` and `4` indicate intermediate quality.
>
> For each clip, we calculated the average score achieved by each method. The summary of average and median scores across all clips is provided in the table below, clearly indicating that our method significantly outperforms the baseline model.
>
> |               | GeneOH Diffusion | TOCH |
> | ------------- | ---------------- | ---- |
> | Average Score | **3.96**         | 1.98 |
> | Median Score  | **4.00**         | 1.55 |
>
> The user study was not initially included due to the extensive visualizations already present in the paper (Sec. 4, Figure 4, 5, Appendix B.1, Figure 8, 9, 10, 11, 12, 13, 14), on the [website](https://geneoh-diffusion.github.io/Geneoh-Diffusion/), and in the [supplementary video](https://geneoh-diffusion.github.io/Geneoh-Diffusion/static/videos/video-2.mp4). The initial quantitative and qualitative evaluations were deemed sufficient to demonstrate the high denoising capability and superiority over previous approaches.
>
> We have since integrated the user study into the revision (*Appendix B.6*).
>
>
>
> *[1] Ye, Y., Hebbar, P., Gupta, A., & Tulsiani, S. (2023). Diffusion-Guided Reconstruction of Everyday Hand-Object Interaction Clips. In Proceedings of the IEEE/CVF International Conference on Computer Vision (pp. 19717-19728).*
>
> *[2] Zhou, K., Bhatnagar, B. L., Lenssen, J. E., & Pons-Moll, G. (2022, October). Toch: Spatio-temporal object-to-hand correspondence for motion refinement. In European Conference on Computer Vision (pp. 1-19). Cham: Springer Nature Switzerland.*
>
> Finally, thank you again for your time and detailed constructive review. We are more than willing to address any further questions and will make every effort to resolve any concerns you may have. Please feel free to let us know if there is anything specific we can provide or clarify.
>
> Looking for your response!
>
> Best regards, Authors

---

> > ### Comment · Reviewer_A9wU · 2023-11-19
> >
> > Dear authors,
> >
> > Thank you for your comprehensive response.
> > * The approach of diffusing an image into Gaussian noise and subsequently denoising it back to an image is a well-established practice in the image generation domain, particularly within diffusion models. It's interesting to note the application of this concept, transitioning from image-to-image generation to its utilization in hand-object interactions.
> >
> > * However, a point of concern is the diffusion sampling process, which requires as many as 700 steps, leading to significantly slow inference times. Could you please clarify the specific method of the diffusion model employed for the reverse denoising process?
> >
> > * Additionally, it would be beneficial if the experimental comparisons included more recent methodologies in this domain. This would provide a clearer context of how your approach stands in comparison to the latest advancements, thereby offering a more comprehensive evaluation of its effectiveness and innovation.

---

> > > ### Author Response · Authors · 2023-11-20
> > > **Reply to A9wU (Part 1 of 2)**
> > >
> > > Dear Reviewer A9wU,
> > >
> > > Thank you for your thoughtful comments. Hope our following responses can adequately address your concerns.
> > >
> > > ### "...diffusing an image into Gaussian noise and subsequently denoising it back to an image is a well-established practice in the image generation domain, particularly within diffusion models. It's interesting..."
> > >
> > > Thank you for your feedback. We would like to underscore the significance of our exploration into the potential of utilizing diffusion techniques to enhance the domain generalization capability of HOI denoising, providing novel insights. Our technical contributions are threefold. Firstly, we introduce a compact and informative representation for representing complex interaction processes (GeneOH, *Sec. 3.1*). Secondly, we present a domain-generalizable denoising scheme that effectively addresses the over-fitting issue that has continuously posed challenges for previous methods (*Sec. 3.2*). Lastly, we contribute a theoretical and practical analysis of critical factors in multi-stage diffusion (*Sec. 3.2*), delving into the nuances of progressive denoising. This includes a discussion on how the multi-stage denoising should be designed to ensure that each stage does not compromise the balance achieved in previous stages. These contributions are crucial to advancing the field.
> > >
> > > ### "...a point of concern is the diffusion sampling process...as many as 700 steps, leading to significantly slow inference times...the specific method of the diffusion model employed for the reverse denoising process?"
> > >
> > > Though designed based on the diffusion process, our inference time is acceptable for many downstream tasks. As mentioned in the previous response, for a sequence with 60 frames, the average inference time is 0.52 seconds for MotionDiff, 16.61 seconds for SpatialDiff, and 7.04 seconds for TemporalDiff. It is because the interaction representation, determined by the number of hand keypoints (21 for the MANO model) and the number of generalized contact points (700 in our experiments), lies in a lower dimensionality than images. Please note that the number of generalized contact points only affects the speed of the SpatialDiff and the TemporalDiff stages.
> > >
> > > Similarly, as pointed out in MDM [1], the inference time for motion generation is acceptable *"since our motion model is small anyway, using dimensions order of magnitude smaller than images"*, though based on diffusion models, requiring 1000 sampling steps,
> > >
> > > Regarding the ***specific method used during inference***, the sampling from the model follows the original iterative approach outlined in DDPM [2]. No advanced inference techniques are utilized.
> > >
> > > In summary, the current inference speed is acceptable for many offline downstream tasks such as animation generation and motion retargeting. We acknowledge that real-time application is not currently achievable. Integrating the advanced techniques developed in works on accelerating diffusion models is a promising research direction. We believe it deserves serious future efforts and exploration.

---

> ### Author Response · Authors · 2023-11-20
> **Reply to A9wU (Part 2 of 2)**
>
> ### "...experimental comparisons included more recent methodologies...a clearer context of how your approach stands in comparison to the latest advancements...effectiveness and innovation."
>
> **Experimental design:** We fully agree with the importance of experimental design, which is why we not only compared our work with the most up-to-date HOI denoising using TOCH but also introduced many *improvements to TOCH methods* and created various ablated versions of our method for comparison, to comprehensively demonstrate the advantages and the positioning of our work. Please note that though we've tried our best to find recent works closely related to ours when preparing the submission, TOCH is the most recent work. However, it cannot tackle the challenging problem to a satisfactory extent. It mainly falls short in removing complicated and heterogeneous interaction noise and suffers from poor generalization ability. Our work wishes to explore a practically useful HOI denoising framework that transcends these shortcomings, thereby expanding the application capabilities of HOI denoising models.
>
> To attain this objective, we propose two main techniques— the progressive multi-stage denoising method and a novel domain-generalizable denoising scheme, diverging from the traditional deterministic mapping-based approach. The design of our diffusion-based denoising scheme, aimed at enhancing domain generalization ability, proves effective and non-trivial, as evident from our comparisons with improved versions of TOCH. The progressive denoising design and the GeneOH representation are instrumental in eliminating heterogeneous and intricate interaction noise, as supported by our thorough ablation studies.
>
> **The interaction denoising problem:** Furthermore, we would like to point out that learning-based denoising represents a relatively nascent area of research, *resulting in a limited number of related works*. After TOCH, it has become challenging to find more recent works for fair comparison through simple improvements, which can be examined from [its citations](https://www.semanticscholar.org/paper/TOCH%3A-Spatio-Temporal-Object-to-Hand-Correspondence-Zhou-Bhatnagar/9c1f78dca7a3ad4a7fee4303958de996c6f34562). The main challenge of this problem, distinct from traditional motion denoising, arises from the intricate spatial-temporal relations between hands and objects during interactions. Merely characterizing these relations poses significant challenges, let alone removing the noise for a clean interaction sequence.
>
> **Comparisons to the latest techniques on post-processing for denoising:** Finally, we tried our best to compare our denoising approach with denoising solutions from recent generative and reconstruction-oriented works (InterDiff [3], ContactGen [4], VisTracker [5]). These works do not necessarily focus on denoising and typically employ *post-processing approaches* following the paradigm of *optimization or deterministic mapping*. Besides, they mainly care about improving the motions based on contacts and assume that there should be contacts established for every frame. However, this is not always true for the general interaction cases we considered in this work, where the hand may not necessarily touch the object in some frames. They are not directly comparable to ours and require modifications for compatibility. We attempted to evaluate the denoising performance using the interaction correction method mentioned in [3] and found that it falls far behind ours (and TOCH) in terms of both performance and generalization. This discrepancy further underscores the unique value of our generalizable denoising network.
>
> *[1] Tevet, G., Raab, S., Gordon, B., Shafir, Y., Cohen-Or, D., & Bermano, A.H. (2022). Human Motion Diffusion Model. ArXiv, abs/2209.14916.*
> *[2] Ho, J., Jain, A., & Abbeel, P. (2020). Denoising Diffusion Probabilistic Models. ArXiv, abs/2006.11239.*
>
> *[3] Xu, S., Li, Z., Wang, Y. X., & Gui, L. Y. (2023). InterDiff: Generating 3D Human-Object Interactions with Physics-Informed Diffusion. In Proceedings of the IEEE/CVF International Conference on Computer Vision (pp. 14928-14940).*
>
> *[4] Liu, S., Zhou, Y., Yang, J., Gupta, S., & Wang, S. (2023). ContactGen: Generative Contact Modeling for Grasp Generation. In Proceedings of the IEEE/CVF International Conference on Computer Vision (pp. 20609-20620).*
>
> *[5] Xie, X., Bhatnagar, B. L., & Pons-Moll, G. (2023). Visibility aware human-object interaction tracking from single rgb camera. In Proceedings of the IEEE/CVF Conference on Computer Vision and Pattern Recognition (pp. 4757-4768).*
>
> Thank you again and looking for your response!
>
> Best regards, Authors

---

> > ### Comment · Reviewer_A9wU · 2023-11-21
> >
> > I am inclined to reconsider my initial evaluation in light of the authors' comprehensive responses to my concerns and the current stagnation in the HOI research domain. The manuscript presents three contributions:
> >
> > * The development of 'GeneOH', a representation that offers a concise yet informative approach to encapsulating complex interaction processes.
> >
> > * The introduction of a domain-generalizable denoising method. This strategy addresses the challenge of over-fitting that has hampered previous methodologies in this area.
> >
> > * An extensive theoretical and practical examination of the nuances in multi-stage diffusion processes.
> >
> > In recognition of these substantial contributions, I believe that this work holds the potential to catalyze advancements in the HOI field.

---

> > > ### Author Response · Authors · 2023-11-21
> > > **Reply to A9wU**
> > >
> > > Thank you!

---

### Official Review · Reviewer_xeyt · 2023-10-30

**Soundness:** 3 good
**Presentation:** 3 good
**Contribution:** 3 good
**Rating:** 6
**Confidence:** 2

**Summary:**

This paper tackles a complex issue - how to remove errors in hand-object interactions to make interaction sequences appear more realistic. This problem involves challenges such as unnatural hand positions, incorrect hand-object relationships, and the need to work well in various interaction scenarios and noise patterns. To address these challenges, the author proposes the framework called GeneOH Diffusion, which includes two key components: GeneOH, a method for representing interactions, and a denoising scheme that works well across different scenarios. The experiments show that the method outperforms others in various tests, suggesting its potential for a wide range of applications.

**Strengths:**

- The writing of this paper is quite good.
- The motivation is clear and the HOI denoising framework is well-designed.
- The experiments are sufficient, where the framework shows promise for novel HOI scenarios.

**Weaknesses:**

I don’t have a big concern with this article since I’m not directly an expert in this area.

**Questions:**

N/A

---

> ### Author Response · Authors · 2023-11-16
> **Author Response**
>
> Dear Reviewer xeyt,
>
> Thank you very much for your time and careful review. We sincerely appreciate your recognition of our challenging problem, the GeneOH design, the effectiveness of our denoising method, the inclusion of sufficient experiments, and the potential value of our method in a wide range of applications.
>
> We are truly grateful for your support. If you have any questions or concerns about our work, please do not hesitate to let us know. We are more than willing to make every effort to address any concerns you may have.
>
> Thank you again, and best regards,
>
> Authors

---

> ### Author Response · Authors · 2023-11-22
> **Grateful for your time and careful review**
>
> Dear Reviewer xeyt,
>
> We do appreciate your recognition and would like to express our sincere gratitude. As we approach the conclusion of the discussion phase, please do not hesitate to let us know if you have any questions. We are committed to trying our best to address your concerns.
>
> Thank you and best regards,
>
> Authors

---

### Official Review · Reviewer_uUah · 2023-10-30

**Soundness:** 4 excellent
**Presentation:** 3 good
**Contribution:** 3 good
**Rating:** 8
**Confidence:** 4

**Summary:**

- The authors introduce GeneOH Diffusion as a solution to address the challenge of denoising generalizable HOIs.
- The authors develop GeneOH, a novel HOI representation that enhances generalization capabilities.
- The authors propose a canonical denoising model for domain-generalizable denoising and a 3-stage progressive approach for HOI denoising.
- This framework holds significant value for various downstream tasks. Experimental results on four test sets showcase the generalization ability of GeneOH Diffusion.

**Strengths:**

- On the technical side, the paper successfully employs diffusion models to achieve domain-generalizable denoising of hand-object interaction.
- The progressive execution of denoising through multiple stages, as mentioned in the article, has merit and aligns with the representation capabilities of GeneOH.
- The author offers a clear definition of a rational hand-object interaction trajectory.
- The paper additionally conducts a comprehensive comparison with various methods using multiple test datasets.

**Weaknesses:**

- The description part of GeneOH can also be further optimized. It is recommended to begin with a summary of the description, including the considered features and the composition of the items. Subsequently, the symbol description can be expanded upon. It is advisable to avoid delving into excessive detail at the initial stages.
- The authors provide experimental evidence demonstrating the generalizability of GeneOH Diffusion to the denoising of hand-object interaction sequences captured by real sensors. However, the majority of the experiments focus on adding known distribution noise to clean ground truth (GT) data and subsequently denoising it. This approach falls short in adequately analyzing the distinction between noise in real hand-object interaction trajectories and artificial noise, as well as evaluating the model's performance in real-world "in-the-wild" scenarios.

**Questions:**

- Some typos: Fig.3 caption: denoosing -> denoising
- Why is it necessary to optimize $J^{stage2}$ using $\mathcal{T}$ to obtain $J^{stage3}$ in the TemporalDiff stage instead of obtaining it through denoising? Have you conducted any ablation experiments to investigate this?
- The authors' ablation experiments demonstrate that removing a certain stage of the diffusion model does not lead to a deterioration in the overall results in terms of IV or Penetration Depth. However, there is a significant degradation observed in HO Motion Consistency. The reviewer requests an explanation from the authors regarding this phenomenon.
- Please consider citing: "H2O: Two Hands Manipulating Objects for First Person Interaction Recognition" in ICCV2021.

---

> ### Author Response · Authors · 2023-11-16
> **Author Response (Part 1 of 2)**
>
> Dear Reviewer uUah,
>
> Thank you so much for your thorough and constructive review. We do appreciate your support and your recognition of the GeneOH representation design, the domain-generalizable denoising scheme, the stage-wise denoising strategy, the comprehensive evaluations, and the demonstrated effectiveness and practical utility of our method.
>
> Below, we address your specific questions in the hope that our responses adequately address any of your concerns.
>
>
>
> ### "...the majority...adding known distribution noise...falls short in...analyzing the distinction between noise in real...and artificial noise...evaluating..."in-the-wild" scenarios."
>
> - We conduct lots of experiments on data with noise from known distributions to better perform ***quantitative evaluations and comparisons*** to previous works. Besides, the *controllability* facilitates us to design a series of test sets for *a systemetic study*.
> - **Testing on real noisy data:** In the meantime, we also place high importance on testing the performance on ***real noisy data***. In the main experiments, we evaluate on the HOI4D dataset. Results are shown in Table 1, Figure 4, 5. More results are included in Table 5, Figure 10, 11 in the appendix. We'd like to emphasize the difficulty of this test setting resulting from the challenging geometry, unobserved articulated motions, and novel large, difficult real noises.
> - **Applications:** In the applications (*Sec. 4.4* and *Appendix B.3*), we underscore our method's practical utility through successful denoising in various real-world scenarios. The nature of the noise in these applications significantly differs from that drawn from known distributions, presenting challenges to prior denoising methods.  We are capable of cleaning them to a satisfactory extent.
> - The real applications currently included could demonstrate the large step we've made to improve the denoising model's application capability.
> - *We are happy to test our model's performance if the reviewer is willing to offer us a concrete setting*.
>
> **Analyzing the distinction between noise in real hand-object interaction trajectories and artificial noise**:
>
> - In the current presentation, the distinctions between real noisy hand trajectories and trajectories with artificial noise are perceptible through visualizations in the paper (Figure 4, 8, 9, 10, 11), the animated results on the [website](https://geneoh-diffusion.github.io/Geneoh-Diffusion/), and the [supplementary video](https://geneoh-diffusion.github.io/Geneoh-Diffusion/static/videos/video-2.mp4). We briefly summarize the differences as follows.
>   - In HOI4D, trajectories consistently exhibit unnatural hand poses, jittering motions, missing contacts, and substantial penetrations.
>
>   - Retargeted hand motions consistently exhibit significant penetrations.
>
>   - Hand trajectories estimated from HOI videos often suffer from penetrations and missing contacts.
>   - A notable characteristic of real noisy hand trajectories is the presence of time-consistent artifacts. Conversely, trajectories with independently added artificial noise to each frame typically manifest time-varying penetrations and unnatural poses.
> - Moreover, as outlined in Table 1, distinctions among different noise patterns are discernible through the analysis of various metrics calculated on their input noisy trajectories, including penetration-related measurements (IV, penetration depth), hand-object proximity (C-IoU, Proximity Error), and motion consistency.
>
> **Further analysis:**
>
> - We extend our analysis by visualizing the differences ($\hat{\mathbf{\theta}} - \mathbf{\theta}^{gt}$) between noisy hand mano pose parameters ($\hat{\mathbf{\theta}}$) and the GT values (${\mathbf{\theta}}^{gt}$) derived from trajectories estimated from videos (the noisy input of the application on cleaning hand trajectory estimations, *Sec. 4.4*), the difference between the mano pose parameters with artificial Gaussian noise ($\hat{\mathbf{\theta}}^{\mathbf{n}}$) and the GT values, along with the differences between the parameters with artificial noise drawn from the Beta distribution ($\hat{\mathbf{\theta}}^{\mathbf{b}}$).
> - We project these differences into a 2-dimensional plane using the PCA algorithm (implemented in the scikit-learn package) and visualize their positions across 256 examples.
> - The figure has been integrated into *Appendix B.5* in the revision. Notably, the noise of the hand trajectory estimated from videos reveals instance-specific patterns in this analysis.
>
> ### "The description part of GeneOH...begin with a summary of the description, ...Subsequently, the symbol description...avoid delving into excessive detail at the initial stages."
>
> Thank you so much for your valuable suggestion! We've incorporated it and revised the overview of GeneOH (*Sec. 3.1*, highlighted in blue). We start with a summary and its considered features, followed by symbols frequently used, and then details of each component.

---

> ### Author Response · Authors · 2023-11-16
> **Author Response (Part 2 of 2)**
>
> ### "...optimize $\mathcal{J}^{\text{stage}2}$ using $\mathcal{T}$ to obtain $\mathcal{J}^{\text{stage}3}$...instead of obtaining it through denoising? ...any ablation experiments...? "
>
> **How does TemporalDiff work?**
>
> - Firstly, we think it is essential to clarify that in both spatial denoising and temporal denoising stages, the model does not directly operate on hand trajectories.
> - Instead, they initially clean their corresponding spatial  $\hat{\mathcal{S}}$ or temporal  $\hat{\mathcal{T}}$ representations of the input respectively. Subsequently, the denoised representation is transformed into the hand trajectory.  The converted trajectory is treated as the denoised result.
> - In the case of SpatialDiff, a straightforward average operation works well and can effectively transform the denoised spatial relations $\mathcal{S}^{\text{stage}2}$ to the denoised $\mathcal{J}^{\text{stage}2}$.
> - However, the transformation of denoised $\mathcal{T}^{\text{stage}3}$ to a hand trajectory poses a more intricate challenge.
>
> **Transforming denoised $\mathcal{T}^{\text{stage}3}$ to three stages-denoised $\mathcal{J}^{\text{stage}3}$: Why we choose an optimization-based strategy?**
>
> - Upon obtaining the denoised temporal relations $\mathcal{T}^{\text{stage}3}$, we proceed to optimize the two-stages denoised $\mathcal{J}^{\text{stage}2}$ to the three-stages denoised $\mathcal{J}^{\text{stage}3}$ using $\mathcal{T}^{\text{stage}3}$. The specific approach is detailed in *Sec. 3.2* and *Appendix A.2*.
>
> - One might question why we opt for an optimization approach rather than decoding relevant information such as HO relative velocity and distances from the denoised representation, followed by conversion to the hand trajectory.
>
> - The rationale lies in the redundancy of temporal information, such as relative velocities, encoded in $\mathcal{T}$. Integrating all decoded information to produce a plausible hand trajectory requires careful consideration, as simple average and integration operations prove impractical.
>
> - In contrast, our optimization-based approach is not only easy to implement but also demonstrates effective performance in our experiments.
>
>
>
>
>
> ### "...removing a certain stage...does not lead to a deterioration in...IV or Penetration Depth...a significant degradation observed in HO Motion Consistency..."
>
> **Phenomenon:** Removing the TemporalDiff stage would leads to large degradation in HO Motion Consistency. However, the IV and Penetration Depth are even better than those of our full model.
>
> **Reason:**
>
> - The removal of the TemporalDiff stage results in observed degradation in HO Motion Consistency since the first two stages do not explicitly model temporal motion. Depending on them alone is insufficient to improve temporal consistency satisfactorily.
> - Missing contacts are a common manifestation of temporal motion inconsistency. In the process of addressing them in the TemporalDiff stage, floating fingers will be attracted to the object surface for correct contact. This process may cause slight penetrations between the hand's skin and the object, resulting in an increase in metrics measuring penetrations.
> - Importantly, no post-processing for removing such slight penetrations is added in the pipeline for quantitative evaluation. Consequently, we observe that the full model exhibits worse performance in penetration-related metrics.
>
> However, we wish to further emphasize that,
>
> - Relying solely on penetration-related metrics to assess the quality of denoised clips is not always reliable. For instance, a scenario where the hand waves in the air without touching the object at all would yield no penetrations, even though it is implausible.
> - Post-processing can effectively remove small penetrations. However, complete removal may not always be necessary. Slight penetrations can be useful in certain downstream tasks, such as simulators that rely on penetrations to generate contact forces. Additionally, they can reflect the soft nature of the hand skin, which deforms when in contact with an object.
>
>
>
> ### "Please consider citing: "H20: Two Hands Manipulating Objects for First Person Interaction Recognition" in ICCV2021."
>
> - Thank you for pointing out this missing reference. We've added it in the revision.
>
>
>
>
> Finally, thank you again for your time and your detailed constructive review. We would love to address any further questions you may have and are committed to resolving any concerns you've raised. Please feel free to let us know if there is additional information we can provide.
>
> Looking forward to your responses!
>
> Best regards, Authors

---

> > ### Comment · Reviewer_uUah · 2023-11-22
> >
> > Dear authors,
> >
> > Thank you for your response.
> > After carefully reading the authors' response and other reviewers' comments, I would like to keep the original rating and am happy to accept it.

---

> > > ### Author Response · Authors · 2023-11-22
> > > **Reply to uUah**
> > >
> > > Thank you! We do appreciate it.

---

### Official Review · Reviewer_C7Zj · 2023-11-03

**Soundness:** 3 good
**Presentation:** 3 good
**Contribution:** 4 excellent
**Rating:** 6
**Confidence:** 3

**Summary:**

This paper proposes a HOI framework with a strong denoising capability to refine the hand trajectories in interaction sequences. The proposed GeneOH Diffusion method has two main key parts, one is a contact-centric HOI representation called GeneOH and the other is a domain-generalizable denoising scheme. Qualitative and Quantitative results show the methods outperform previous methods.

**Strengths:**

- The proposed method has strong spatial and temporal denoising capability and it generalizes well to novel HOI scenes. Other than existing methods, this paper explicitly parameterizes both the spatial and temporal relations between hands and objects.
- The proposed progressive HOI denoising process is designed to use a separate stage to learn hand trajectory, and hand-object spatial and temporal relations collaboratively.
- The proposed method outperforms the previous method TOCH on all 4 test sets by a large margin.

**Weaknesses:**

- In Fig 5 last column middle row, when the hand manipulating the scissors, the grasp does not make sense and the hand shown is penetrated into the scissors.
- The method has the assumption that it requires accurate object pose trajectory to work.

**Questions:**

- In Sec 3.2 Fitting for a hand mesh trajectory, the method denoises hand key points in the diffusion process and in the end optimizes to MANO parameters. How does the method ensure there is no penetration after going from keypoints to MANO mesh?
- In Sec 4.1 Training dataset, why did the authors use different standard deviations to noise the MANO parameters for translation, rotation, and pose parameters Does the way to noise the training sets affect the learning?
- In Sec 4.1 Evaluation dataset, what is the reason to use different noise on different test sets? Hope the authors give more explanation about it.

---

> ### Author Response · Authors · 2023-11-16
> **Author Response (Part 1 of 3)**
>
> Dear Reviewer C7Zi,
>
> Thank you so much for your detailed and constructive review. We appreciate your recognition of the GeneOH representation design, the stage-wise denoising, the domain-generalizable denoising scheme, and the high denoising ability of our method, as well as its superiority over previous approaches. In the following text, we provide our responses to your questions. Hope they can address your concerns.
>
>
>
> ### "...assumption...requires accurate object pose trajectory to work."
>
> - **Wide application scenarios:** Denoising hand trajectories with accurate object poses is an important setting with many practical demands in applications, such as motion retargeting [3] and virtual object manipulation [1,2]. Additionally, some HOI datasets are constructed by tracking accurate object poses through MoCap while reconstructing hand movements via vision estimation pipelines. In these cases, the object poses are accurate in the data.
> - **Our high practical utility:** Our method is a practically useful tool in many applications, including refining motion-retargeted trajectories (*Sec. 4.4*), cleaning estimations from videos (*Sec. 4.4*), cleaning the real noisy dataset (main experiments on HOI4D, *Sec. 4.2*), refining synthesized grasps (Appendix *B.3*), and serving as a data creation tool for synthesis networks (Appendix *B.3*).
> - **Difficult problem and our contributions:** Generalizable HOI denoising is inherently challenging, even with the assumption of accurate object poses. Our work represents a significant advancement, addressing a previously unsolvable problem to a satisfactory level—a substantial step forward compared to previous approaches.
>
> *[1] Oh, J. Y., Park, J. H., & Park, J. M. (2019). Virtual object manipulation by combining touch and head interactions for mobile augmented reality. Applied Sciences, 9(14), 2933.*
>
> *[2] Shaer, O., & Hornecker, E. (2010). Tangible user interfaces: past, present, and future directions. Foundations and Trends® in Human–Computer Intera*
>
> *[3] Aberman, K., Wu, R., Lischinski, D., Chen, B., & Cohen-Or, D. (2019). Learning character-agnostic motion for motion retargeting in 2d. arXiv preprint arXiv:1905.01680.*
>
>
>
> ### "...denoises hand key points...no penetration after going from keypoints to MANO mesh?"
>
> - **Learning priors of non-penetrating meshes from keypoints:** The distribution of spatial relations between non-penetrating meshes can be partially modeled by the distribution of natural spatial relations between keypoints and the object.
>   - Large penetrations, such as a portion of a finger penetrating through the object, can be explicitly described by the spatial relations between hand keypoints and the object.
>   - Small penetrations can be revealed by considering the distance between hand keypoints and the object surface. The hand keypoints, chosen as hand skeleton points in our method, should maintain a certain distance from the object, and this distance distribution can be learned by the model from the clean interaction data.
> - **Elimination of small penetrations:** Simple post-processing techniques, such as penalizing hand-object penetration volumes, can effectively eliminate unavoidable small penetrations.
> - **Slight penetrations hold utility and can reflect natural soft hand deformations**:
>   - Some simulators with penalty-based contact models rely on penetrations to generate contact forces for interacting objects (like *Drake (simple version) [3], DiffRedMax [1,2]*).
>   - Furthermore, these penetrations can accurately depict realistic soft hand deformations during interactions with objects, as stated in *ContactOpt [4]*.
>
> *[1] Xu, J., Chen, T., Zlokapa, L., Foshey, M., Matusik, W., Sueda, S., & Agrawal, P. (2021). An end-to-end differentiable framework for contact-aware robot design. arXiv preprint arXiv:2107.07501.*
>
> *[2] Wang, Y., Weidner, N. J., Baxter, M. A., Hwang, Y., Kaufman, D. M., & Sueda, S. (2019). REDMAX: Efficient & flexible approach for articulated dynamics. ACM Transactions on Graphics (TOG), 38(4), 1-10.*
>
> *[3] Tedrake, R. (2019). Drake: Model-based design and verification for robotics.*
>
> *[4] Grady, P., Tang, C., Twigg, C. D., Vo, M., Brahmbhatt, S., & Kemp, C. C. (2021). Contactopt: Optimizing contact to improve grasps. In Proceedings of the IEEE/CVF Conference on Computer Vision and Pattern Recognition (pp. 1471-1481).*

---

> ### Author Response · Authors · 2023-11-16
> **Author Response (Part 2 of 3)**
>
> ### "Fig 5...the scissors, the grasp does not make sense and the hand shown is penetrated into the scissors."
>
> - In this case, where the hand waves above the scissor, failing to establish sufficient contacts to manipulate the object, noise removal from the input sequence becomes ambiguous.
>
> - Our method offers two plausible approaches to refine the trajectory:1) Putting the thumb out of the scissor's ring (*the middle row*), and 2) Inserting it into the circle (*the third row*). Both of them are plausible since they can provide enough contacts to manipulate the object.
>
> - Remaining slight penetrations between the hand skin and the object can be further cleaned through simple post-processing, although this is not included in our default method. However, as clarified in our response to the previous question, fully cleaning these slight penetrations is often unnecessary. They can reflect natural soft deformations and are useful in various applications.
>
>
>
> ### "...Training...different standard deviations to noise the MANO parameters...Does the way to noise...affect the learning?"
>
> - Noise scales for different parameters diverge due to two main reasons: 1) The inherent values of these parameters are at different scales. 2) Hand global transformations are more accurately tracked in real scenarios. So we keep the noise scales added on the rotation and translation parameters to relatively small values accordingly to align with real situations.
>
> - It's worth noting that the way noise is added would only impact the learning of deterministic denoising methods, such as the baseline TOCH [1] and our ablated version "Ours w/o Diffusion". However, our full model, incorporating a stochastic denoising paradigm, would not be influenced.
>
> - For deterministic methods, the current noise addition approach aligns with the methodology used in constructing the GRAB test set. This ensures that the denoising model, trained on this dataset, encounters minimal domain shift when tested on the GRAB test set. If a different trend of noise scales were used during training, the performance of deterministic methods on the easiest GRAB test set would get worse.
>
> *[1] Zhou, K., Bhatnagar, B. L., Lenssen, J. E., & Pons-Moll, G. (2022, October). Toch: Spatio-temporal object-to-hand correspondence for motion refinement. In European Conference on Computer Vision (pp. 1-19). Cham: Springer Nature Switzerland.*

---

> ### Author Response · Authors · 2023-11-16
> **Author Response (Part 3 of 3)**
>
> ### "...Evaluation...what is the reason to use different noise on different test sets?"
>
> We are not sure if the reviewer's question is *a)* why we use different noise on different test sets, rather than using the same noise, or *b)* why we use different noise on different test sets instead of noising each of them with each possible noise. Please allow us to provide explanations for both *a)* and *b)*.
>
> - The reason for *a)* is to construct a series of test sets with varying levels of domain shifts from the training set, assessing the model's denoising capability across a spectrum of scenarios, ranging from the easiest with only new objects and unseen interactions to the most challenging scenarios closely aligned with real applications.
> - In real-world applications, the noise pattern in the input often differs from that observed during training and cannot be precisely described by a specific distribution with a clear mathematical formulation. Generalizing to unseen noise distributions is challenging but necessary for a denoising model to be applicable in real settings.
> - To systematically evaluate the robustness of denoising models to different kinds of noise distribution shifts, we set up a series of test sets with varying levels of noise distribution shifts—ranging from the noise distribution identical to the training dataset to an unseen synthetic noise distribution, and finally to novel real noise patterns.
> - In the following table, we introduce the evaluation setting, focusing on the domain shifts of each test set from the training dataset.
>
> |                      | New Objects                         | New Interactions                                | Novel Noise     | Evaluation Purpose                                           |
> | -------------------- | ----------------------------------- | ----------------------------------------------- | --------------- | ------------------------------------------------------------ |
> | GRAB test set        | Yes                                 | Yes                                             | No              | Generalization ability towards unseen interactions with new objects |
> | ARCTIC dataset       | Yes (including articulated objects) | Yes (bimanual manipulations; changing contacts) | No              | Generalization ability towards unseen difficult interactions with new objects |
> | GRAB (Beta) test set | Yes                                 | Yes                                             | Yes (synthetic) | Generalization ability towards ***novel synthetic interaction noise*** and unseen interactions with new objects |
> | HOI4D dataset        | Yes (including articulated objects) | Yes                                             | Yes (real)      | Generalization ability towards ***novel, challenging, real interaction noise*** and unseen interactions with new objects |
>
> If the question is *b)*, we'd like to further explain that
>
> - The default noising strategy involves adding Gaussian noise with noise scales identical to those used in the training dataset. Consequently, for both the GRAB and ARCTIC datasets, their default test sets are constructed using Gaussian noise.
> - To further assess the model's generalization ability to novel synthetic noise distributions, we introduce GRAB (Beta), which employs a different noising strategy. In the meantime, the objects and interactions are controlled to be the same as the GRAB test set.
> - ARCTIC is primarily introduced to evaluate the model's generalization ability to unseen and challenging interactions.
> - Furthermore, the HOI4D dataset provides noisy trajectories with real noise patterns. However, these real noise patterns cannot be distilled into a specific noising strategy to noise other datasets. Therefore, the model's robustness towards real noise can only be tested on this specific dataset.
>
>
>
> At last, thank you again for your time and your detailed constructive review. We would love to answer any further questions and will do our best to address any concerns you may have. Please let us know if there is anything specific we can provide.
>
> Looking forward to your responses!
>
> Best regards, Authors

---

> ### Author Response · Authors · 2023-11-22
> **Thank you for your careful review and look forward to post-rebuttal feedbacks**
>
> Dear Reviewer C7Zj,
>
> Thank you again for your valuable review and insightful comments.
>
> We have provided comprehensive responses above regarding your questions on the problem setting, concerns about avoiding penetrations, and inquiries into the noising strategies employed during the training and the evaluation. We also clarified the difficulty of the problem, our contributions, and the high practical value, as well as the rationale of our designs on the denoising method, the noising strategies, and the experimental setting.
>
> As the end of the discussion phase is approaching, we are wondering whether your questions and concerns have been thoroughly addressed. If there is any additional information or clarification that we can offer, please do not hesitate to let us know. We’ll spare no effort in addressing any of your remaining concerns. We are happy to have further discussions!
>
> Best regards,
>
> Authors

---

### Author Response · Authors · 2023-11-18
**Thank you for all your insightful comments and look forward to post-rebuttal feedbacks**

Dear Reviewers,

We'd like to express our sincere gratitude for your detailed reviews and valuable suggestions.

We have provided responses to all of your questions and have incorporated your suggestions in the revised paper (highlighted in blue). We hope our explanations and revisions could address your concerns.

As the discussion phase will conclude on November 22nd, please don’t hesitate to let us know if you have any further questions. We are committed to exerting every effort to address any remaining concerns you may have. We appreciate your suggestions.

Thank you for your time and constructive reviews! Sincerely look forward to your responses.

Best regards,

Authors

---

### Meta-Review · Area_Chair_r2Lz · 2023-12-09

**Metareview:**

The paper was reviewed by 4 experts, and initially had positive reviews (8, 6, 6, 6). The major concerns were:

1. missing explanations/justifications about the methodology and evaluation.
2. missing evaluation of "in the wild" scenes with real noise?
3. complicated framework.
4. missing some recent works.
5. incremental method: reusing the diffusion framework.
6. no user study.

The authors wrote a response to address the concerns. Specifically, they added more explanations/justifications and pointed out the real-world noise experiments. They explained their contributions required to perform HOI denoting with diffusion, as well as the difference with related works, and conducted a user study. All reviewers were positive on the paper, and appreciated the solution making HOI generalizable denoising, good evaluation performance, and the potential impact on downstream tasks.

The AC agrees with the reviewers and recommends accept. The authors should prepare a camera-version based on the reviews and discussion.

**Justification For Why Not Higher Score:**

It's an application paper. The core ML part is the diffusion model, applied progressively to remove different types of noise.

**Justification For Why Not Lower Score:**

The results are good, and there is novelty in the proposed methodology specific to HOI for using diffusion-based denoising. The good results can impact downstream applications. Furthermore, this successful application could inspire others to use diffusion models in non-typical ways.

---

### Decision · Program_Chairs · 2024-01-16

Accept (poster)